# ROBUST MIXTURE MODELS FOR ALGORITHMIC FAIRNESS UNDER LATENT HETEROGENEITY

## ABSTRACT

Standard machine learning models optimized for average performance often fail on minority subgroups and lack robustness to distribution shifts. This challenge worsens when subgroups are latent and affected by complex interactions among continuous and discrete features. We introduce **ROME** (**RO**bust **M**ixture **E**nsemble), a framework that learns latent group structure from data while optimizing for worst-group performance. ROME employs two approaches: an Expectation-Maximization algorithm for linear models and a neural Mixture-of-Experts for nonlinear settings. Through simulations and experiments on real-world datasets, we demonstrate that ROME significantly improves algorithmic fairness compared to standard methods while maintaining competitive average performance. Importantly, our method requires no predefined group labels, making it practical when sources of disparities are unknown or evolving. Implementations in R and Python are available at this anonymous repository.

## 1 INTRODUCTION

Deploying machine learning models in high-stakes domains such as healthcare (Ning et al., 2024), criminal justice (Ávila et al., 2020), and finance (Das et al., 2021) requires both accuracy and equity. However, models optimized for average performance often fail on minority subpopulations (Barocas & Selbst, 2016). Most existing fairness methods (Li et al., 2025; Cui et al., 2021) focus on disparities across predefined, discrete demographic groups (e.g., binary gender, racial categories) (Mitchell et al., 2021). These approaches have two fundamental limitations: (1) They assume bias manifests along observable groups, and (2) they cannot handle continuous attributes like income or age without arbitrary discretizations (Shilova et al., 2025).

These limitations motivate our focus on discovering latent groups—subpopulations defined by complex interactions between features that standard demographic categories miss. Classical mixture models provide a foundation for modeling heterogeneous populations with $G$ unobserved subgroups (McLachlan, 2000). For observed data $\boldsymbol{y}$, the marginal density is:

$$f(\boldsymbol{y}) = \sum_{g=1}^{G} \pi_g f_g(\boldsymbol{y}),$$

where $\pi_g \geq 0$ are mixing proportions ($\sum_g \pi_g = 1$) and $f_g(\boldsymbol{y})$ is the component density for group $g$. Group membership is represented by latent indicators $z_{ig} \in \{0, 1\}$, where $z_{ig} = 1$ if observation $i$ belongs to group $g$. These models are typically estimated via Expectation-Maximization (Dempster et al., 1977). In deep learning, Mixture of Experts (MoE) architectures (Jordan & Jacobs, 1994; Shazeer et al., 2017) implement similar principles through gating networks that route inputs to specialized experts. While MoE traditionally focuses on computational efficiency and capacity, we re-purpose these architectures for algorithmic fairness.

Beyond latent group discovery, we must ensure robustness. Under distribution shift (Yang et al., 2022), performance degrades most for vulnerable groups, yet standard empirical risk minimization allows arbitrary

degradation on minorities while maintaining high average accuracy. Distributionally robust optimization (DRO) addresses this by optimizing worst-case performance over uncertainty sets (Sagawa et al., 2020). Wang et al. (2023) showed that for multi-source data with known group structure, the distributionally robust predictor admits a closed-form solution as a weighted average of source-specific models. We extend this framework to settings where groups are latent and must be discovered from data.

**Our Contributions:** We propose **ROME** (**RO**bust **M**ixture **E**nsemble) to handle algorithmic disparities in settings involving latent, unobserved groups related to both continuous and discrete features. Our key insight is that the latent groups discovered by mixture models can serve as the source populations in the DRO framework, with the optimal aggregation weights providing interpretable measures of each latent group's contribution. Through both simulation studies and real-world data, we provide a comparative analysis of these approaches, highlighting key trade-offs and implications for fair, robust prediction for decision-making.

## 2 ROME-EM FOR LINEAR MODELS

### 2.1 PROBLEM SETUP

We consider a supervised learning problem where each observation consists of an outcome $Y_i \in \mathbb{R}$ and features $\boldsymbol{X}_i = (\boldsymbol{A}_i, \boldsymbol{S}_i)$. The vector $\boldsymbol{A}_i \in \mathbb{R}^{p_A}$ contains non-sensitive features (e.g., clinical measurements), while $\boldsymbol{S}_i = (S_{i1}, \ldots, S_{ip_S}) \in \mathbb{R}^{p_S}$ contains sensitive attributes (e.g., demographics, socioeconomic status). We assume the population consists of $G$ latent groups. Let $z_{ij} \in \{0, 1\}$ denote the latent group indicator where $z_{ij} = 1$ if observation $i$ belongs to group $j$. The membership probabilities follow a multinomial logistic model using selected sensitive attributes $\boldsymbol{S}_{i,\text{mem}} = \{S_{ik} : k \in \mathcal{I}_{\text{mem}}\}$

$$P(z_{ij} = 1 \mid \boldsymbol{S}_{i,\text{mem}}) = \frac{\exp(\boldsymbol{\gamma}_j^\top \boldsymbol{S}_{i,\text{mem}})}{\sum_{k=1}^G \exp(\boldsymbol{\gamma}_k^\top \boldsymbol{S}_{i,\text{mem}})} \tag{1}$$

where $\mathcal{I}_{\text{mem}} \subseteq \{1, \cdots, p_S\}$ denotes the indices of sensitive features used for group membership model, and $\boldsymbol{\gamma}_j \in \mathbb{R}^{|\mathcal{I}_{\text{mem}}|}$ are group-specific membership parameters. Given group membership, the continuous outcome follows a group-specific linear model:

$$Y_i \mid z_{ij} \sim \mathcal{N}(\boldsymbol{\omega}_j^\top \boldsymbol{X}_i, \sigma^2) \tag{2}$$

where $\boldsymbol{X}_i = [1, \boldsymbol{A}_i, \boldsymbol{S}_{i,\text{out}}]$ is the design vector with $\boldsymbol{S}_{i,\text{out}} = \{S_{ik} : k \in \mathcal{I}_{\text{out}}\}$ being the sensitive features included in outcome prediction model and $\boldsymbol{\omega}_j \in \mathbb{R}^{1+p_A+|\mathcal{I}_{\text{out}}|}$. The feature selection indices $\mathcal{I}_{\text{mem}}$ and $\mathcal{I}_{\text{out}}$ enable flexible modeling based on domain constraints. For instance, we might use all demographic features to identify groups while excluding protected attributes from outcome prediction. Throughout this work we distinguish two roles of sensitive attributes: $S_{i,\text{mem}}$ variables allowed to inform the estimation of latent subgroup structure; and $S_{i,\text{out}}$ variables permitted to enter the final outcome prediction model[1].

**Fairness notion.** In this work, we adopt a structural, group-robust view of fairness: a model is considered fair if its predictive performance is stable across subpopulations. This notion is particularly suitable for settings with continuous outcomes, where many fairness definitions developed for binary classification (e.g., demographic parity, equalized odds) (Bird et al., 2020; Hong et al., 2024; Du et al., 2021) do not extend naturally or lead to ill-defined interpretations. Accordingly, we focus on group-wise risk measures as our primary fairness criteria. Smaller gaps between the best- and worst-performing subgroups, together with low overall risk, correspond to improved group fairness in our setting.

### 2.2 EM ALGORITHM

Since group labels are unobserved, we use the expectation maximization (EM) algorithm to estimate parameters $\boldsymbol{\Theta} = \{\boldsymbol{\gamma}_1, \ldots, \boldsymbol{\gamma}_G, \boldsymbol{\omega}_1, \ldots, \boldsymbol{\omega}_G\}$.

---

[1] The determination of which attributes belong to $S_{i,\text{mem}}$ and $S_{i,\text{out}}$ is entirely guided by domain knowledge, institutional guidelines, and fairness/compliance requirements, not by the method itself.

**Initialization:** The initial hard assignments $\{z_{ij}^{(0)}\}_{i=1}^n$ provide a warm start for EM and are replaced by soft probabilities in subsequent iterations. They can be either: (1) provided based on domain knowledge (e.g., using sensitive attributes as input of clustering methods to obtain initial hard assignments), or (2) randomly sampled from $\text{Uniform}(1, G)$. For membership parameters $\boldsymbol{\gamma}_j$, we fit a logistic regression of the initial group indicators on $\boldsymbol{S}_{\text{mem}}$. For outcome parameters $\boldsymbol{\omega}_j$, we fit group-specific linear regressions when $n_j$ is sufficiently large (Ng et al., 2011; Levine & Casella, 2001); otherwise, we use pooled regression estimates to ensure numerical stability.

**E-Step:** Given current parameters $\{\boldsymbol{\gamma}_j^{(t)}, \boldsymbol{\omega}_j^{(t)}\}_{j=1}^G$, we compute the posterior probabilities:

$$w_{ij}^{(t)} = \frac{p_{ij}^{(t)} \cdot \ell_{ij}^{(t)}}{\sum_{k=1}^G p_{ik}^{(t)} \cdot \ell_{ik}^{(t)}},$$

where $p_{ij}^{(t)} = P(z_{ij} = 1 \mid \boldsymbol{S}_{i,\text{mem}}; \boldsymbol{\gamma}_j^{(t)})$ is the membership probability of observation $i$ under group $j$ and

$$\ell_{ij}^{(t)} = \exp\left(-\frac{1}{2}(Y_i - \boldsymbol{\omega}_j^{(t)\top}\boldsymbol{X}_i)^2\right)$$

is the Gaussian likelihood, with fixed variance absorbed into the proportionality.

**M-Step with Backtracking Line Search:** To ensure monotonic increase in the log-likelihood, we employ backtracking line search (Liang, 2021) for parameter updates. For each parameter update from $\boldsymbol{\theta}^{(t)}$ to $\boldsymbol{\theta}^{(t+1)}$, we set: $\boldsymbol{\theta}^{(t+1)} = \boldsymbol{\theta}^{(t)} + \alpha(\boldsymbol{\theta}_{\text{new}} - \boldsymbol{\theta}^{(t)})$ starting with $\alpha = 0.5$ and halving $\alpha$ until the likelihood improves or $\alpha < \tau_2$. For each group, we update $\boldsymbol{\gamma}_j$ via weighted logistic regression with weights $w_{ij}^{(t)}$, using a quasi-binomial family to handle potential overdispersion:

$$\boldsymbol{\gamma}_j^{\text{new}} = \arg\max_{\boldsymbol{\gamma}} \sum_{i=1}^n w_{ij}^{(t)} \left[\boldsymbol{\gamma}^\top \boldsymbol{S}_{i,\text{mem}} - \log\left(\sum_{k=1}^G \exp(\boldsymbol{\gamma}_k^\top \boldsymbol{S}_{i,\text{mem}})\right)\right]$$

We solve the weighted least squares problem: $\boldsymbol{\omega}_j^{\text{new}} = (\boldsymbol{X}^\top \boldsymbol{W}_j^{(t)} \boldsymbol{X})^{-1} \boldsymbol{X}^\top \boldsymbol{W}_j^{(t)} \boldsymbol{Y}$, where $\boldsymbol{W}_j^{(t)} = \text{diag}(w_{1j}^{(t)}, \ldots, w_{nj}^{(t)})$.

**Convergence and Model Selection:** The algorithm terminates when the total parameter change falls below tolerance $\tau_1$:

$$\sum_{j=1}^G \left(||\boldsymbol{\gamma}_j^{(t+1)} - \boldsymbol{\gamma}_j^{(t)}||_1 + ||\boldsymbol{\omega}_j^{(t+1)} - \boldsymbol{\omega}_j^{(t)}||_1\right) < \tau_1$$

or maximum iterations are reached. In practice, the number of groups $G$ could be fine-tuned using information criteria (AIC or BIC) computed from the maximized log-likelihood.

## 2.3 DRO Aggregation

After obtaining group-specific models $\{\hat{\boldsymbol{\omega}}_1, \ldots, \hat{\boldsymbol{\omega}}_G\}$ from the EM, we employ a distributionally robust optimization (DRO) framework to construct a robust predictor that performs well across different mixture distributions of the latent groups.

**DRO Formulation.** We consider mixture distributions of the $G$ group-specific conditionals:

$$\mathbb{P}_{Y|X}^q = \sum_{j=1}^G q_j\, \mathbb{P}_{Y|X}^{(j)}, \qquad q \in \mathcal{H},$$

where $\mathcal{H} = \left\{ \mathbf{v} \in \Delta^G : \|\mathbf{v} - \mathbf{v}_0\|_2 \leq c\sqrt{G} \right\} \subseteq \Delta^G$ is an uncertainty set over mixture weights. Here $\mathbf{v}_0$ represents a baseline weight (default: uniform weights $(1/G, \ldots, 1/G)$ and constant $c \in [0, 1]$ controls the size of the uncertainty region. The distributionally robust predictor solves

$$f^* = \arg\max_{f \in \mathcal{F}} \min_{q \in \mathcal{H}} R_q(f), f \in \mathcal{F}$$

where $R_q(f)$ denotes the explained variance under mixture weights $q$.

**Closed-Form Solution:** Following key results from Wang et al. (2023), when $\mathcal{F}$ consists of convex combinations of the group-specific estimated predictors $\hat{f}_j(\boldsymbol{X}) = \hat{\boldsymbol{\omega}}_j^\top \boldsymbol{X}$, the robust predictor $f_{\boldsymbol{v}}$ has the closed form:

$$f_{\boldsymbol{v}}(\boldsymbol{X}_i) = \sum_{j=1}^{G} v_j \cdot \hat{\boldsymbol{\omega}}_j^\top \boldsymbol{X}_i \tag{3}$$

where the optimal weights solve:

$$\boldsymbol{v}^* = \arg\min_{v \in \mathcal{H}} \boldsymbol{v}^\top \hat{\boldsymbol{\Gamma}} \boldsymbol{v} \tag{4}$$

Here $\boldsymbol{\Gamma}$ is a $G \times G$ matrix, representing the empirical covariance of predictions across groups, with entries $\boldsymbol{\Gamma}_{k,j} = \mathbb{E}_{Q_X}[f^{(k)}(X)f^{(j)}(X)]$ and we compute $\hat{\boldsymbol{\Gamma}}$ by $\hat{\boldsymbol{\Gamma}}_{jk} = \frac{1}{n}\sum_{i=1}^{n} \hat{f}_j(\boldsymbol{X}_i) \cdot \hat{f}_k(\boldsymbol{X}_i)$.

**Final Robust Predictor:** Finally, the robust predictor of ROME-EM for $\boldsymbol{X}_{\text{new}}$ is:

$$\hat{Y}_{\text{ROME-EM}} = \sum_{j=1}^{G} v_j^* \cdot \hat{\boldsymbol{\omega}}_j^\top \boldsymbol{X}_{\text{new}} \tag{5}$$

Our formulation leverages the closed-form maximin aggregation structure derived in the linear setting, which provides the key theoretical motivation for incorporating robust group-wise risk control into mixture and MoE architectures.

# 3 ROME-MoE FOR NONLINEAR MODELS

While ROME-EM follows a classical EM-style update structure and includes a closed-form robust aggregation step under mixture model assumptions, the linear multinomial assumption for group membership (Equation 1) makes ROME-EM vulnerable to misspecification when true group structures involve non-linear or complex feature interactions.

Though neural network EM variants exist (Nagpal et al., 2022), they sacrifice the closed-form DRO solution and convergence guarantees that make ROME-EM theoretically appealing. We therefore pursue a different approach: extending ROME to neural networks using Mixture of Experts (MoE) (Shazeer et al., 2017), which naturally handles non-linear relationships while incorporating DRO principles directly into the training objective. Specifically, ROME-MoE treats experts as group-specific predictors and uses a loss function that explicitly balances average and worst-group performance, operationalizing the maximin principle from Section 2.3 in an end-to-end differentiable framework.

Our ROME-MoE architecture consists of two key components: (1) **Gating network:** A neural network $g : \mathbb{R}^p \to \Delta^G$ which maps features of dimension $p$ (could be decided flexibly by the user via domain knowledge), to a probability distribution over $G$ experts, determining soft group assignments. (2) **Expert network:** A collection of $G$ neural networks $\{f_1, \ldots, f_G\}$ where each $f_j : \mathbb{R}^{p_A} \to \mathbb{R}$ specializes in predictions for its corresponding latent group using only non-sensitive features. We present two variants: ROME-MoE-S uses only sensitive features $\boldsymbol{S}$ for the gating network (assuming group membership is primarily determined by $\boldsymbol{S}$), while ROME-MoE-AS uses all features. The key fairness constraint in ROME-MoE is architectural: $\boldsymbol{S}$ can influence group assignment through the gating network but cannot directly affect predictions, as experts only access non-sensitive features $\boldsymbol{A}$.

---

**Algorithm 1** ROME-MoE

---

**Require:** Training data $\mathcal{D} = \{(\boldsymbol{A}_i, \boldsymbol{S}_i, Y_i)\}_{i=1}^n$, number of experts $G$, DRO parameter $\alpha \in [0, 1]$, learning rate $\eta$, batch size $B$, epochs $E$

**Ensure:** Trained ROME-MoE model with parameters $\Theta = \{\theta_{\text{gate}}, \theta_1, \ldots, \theta_G\}$

1: **Initialize:** Gating network $g_{\theta_{\text{gate}}}$, Expert networks $\{f_{\theta_1}, \ldots, f_{\theta_G}\}$
2: **for** epoch $= 1$ to $E$ **do**
3:     **for** each batch $\mathcal{B} \subset \mathcal{D}$ with $|\mathcal{B}| = B$ **do**
4:         **for** each $(\boldsymbol{A}_i, \boldsymbol{S}_i, Y_i) \in \mathcal{B}$ **do**
5:             **if** ROME−MoE−S **then**
6:                 $\boldsymbol{w}_i = \text{Softmax}(g_{\theta_{\text{gate}}}(\boldsymbol{S}_i))$                   $\triangleright$ ROME−MoE−S
7:             **else**                                            $\triangleright$ ROME−MoE−AS
8:                 $\boldsymbol{w}_i = \text{Softmax}(g_{\theta_{\text{gate}}}([\boldsymbol{A}_i; \boldsymbol{S}_i]))$
9:             **end if**
10:             $\hat{Y}_i = \sum_{j=1}^G w_{ij} \cdot f_{\theta_j}(\boldsymbol{A}_i)$              $\triangleright$ Weighted expert predictions
11:         **end for**
12:         **for** $j = 1$ to $G$ **do**
13:             $\mathcal{I}_j = \{i \in \mathcal{B} : w_{ij} > 0.1\}$             $\triangleright$ Samples with significant membership
14:             **if** $|\mathcal{I}_j| > 0$ **then**
15:                 $\mathcal{L}_j = \frac{1}{|\mathcal{I}_j|} \sum_{i \in \mathcal{I}_j} w_{ij} \cdot (Y_i - \hat{Y}_i)^2$
16:             **else**
17:                 $\mathcal{L}_j = 0$
18:             **end if**
19:         **end for**
20:         **// DRO Objective**
21:         $\mathcal{L}_{\text{avg}} = \frac{1}{B} \sum_{i \in \mathcal{B}} (Y_i - \hat{Y}_i)^2$                 $\triangleright$ Average loss
22:         $\mathcal{L}_{\text{worst}} = \max_{j \in \{1, \ldots, G\}} \mathcal{L}_j$             $\triangleright$ Worst group loss
23:         $\mathcal{L}_{\text{total}} = (1 - \alpha) \cdot \mathcal{L}_{\text{avg}} + \alpha \cdot \mathcal{L}_{\text{worst}}$
24:         **// Backward Pass**
25:         Compute gradients: $\nabla_\Theta \mathcal{L}_{\text{total}}$
26:         Update parameters: $\Theta \leftarrow \Theta - \eta \cdot \nabla_\Theta \mathcal{L}_{\text{total}}$
27:     **end for**
28: **end for**
29: **return** Trained model with parameters $\Theta$

---

**Fair-use of sensitive attributes.** We emphasize that sensitive features $S$ are used only inside the gating network in order to estimate latent subgroup structure; the experts never receive $S$. Thus, $S$ influences which expert is selected, but does not enter the prediction function itself. This is the standard structure in latent-group fairness and mixture modeling, and is analogous to the role of $S$ in ROME-EM, where it is used only for membership inference. Users specify which features belong to $S$ based on policy or legal constraints, ensuring that the final predictor does not depend on those fairness-sensitive variables.

The detailed implementation is available in Algorithm 1. Unlike Vanilla MoE training that minimizes average loss, ROME-MoE incorporates distributionally robust optimization directly into the training objective. Given a batch of data, we compute group assignments through the gating network and optimize:

$$\mathcal{L}_{\text{ROME-MoE}} = (1 - \alpha) \cdot \mathcal{L}_{\text{avg}} + \alpha \cdot \mathcal{L}_{\text{worst}} \tag{6}$$

where $\mathcal{L}_{\text{avg}}$ is the average loss across all samples, $\mathcal{L}_{\text{worst}} = \max_{j \in \{1, \ldots, G\}} \mathcal{L}_j$ is the worst group loss, and $\alpha \in (0, 1)$ controls the trade-off between average and worst-case performance. In practice, we found $\alpha \in [0.05, 0.1]$ provides a good balance between average and worst-group performance.

## 4 EXPERIMENTS

### 4.1 SIMULATION STUDY

**Data Generation and Setup** We conducted a comprehensive simulation study to evaluate ROME-EM's ability to recover latent group structure and improve worst-group performance. We generated data with $n = 8,000$ observations from $G = 4$ latent groups, each with distinct parameter vectors and mixing proportions. Group membership was determined by a multinomial logistic model based on sensitive attributes $\boldsymbol{S} \in \mathbb{R}^5$, with each group having distinct membership parameters. For the outcome model, we generated $p = 20$ total features comprising $p_A = 15$ non-sensitive features and $p_S = 5$ sensitive attributes.

The true regression coefficients $\beta_j$ for each group were constructed using a scaled perturbation approach to introduce meaningful heterogeneity across groups while maintaining realistic parameter magnitudes. Complete simulation parameters are provided in Appendix A with details of in Appendix A.1. To assess robustness to initialization, we intentionally misspecified the starting conditions by randomly reassigning 50% of observations to incorrect groups before running ROME-EM.

For the DRO optimization stage, we evaluated multiple constraint values to identify the optimal fairness-efficiency trade-off (detailed in Appendix A). Each of the 100 simulation replications was evaluated on an independent test set of $n = 8,000$ observations generated from the same underlying distributions.

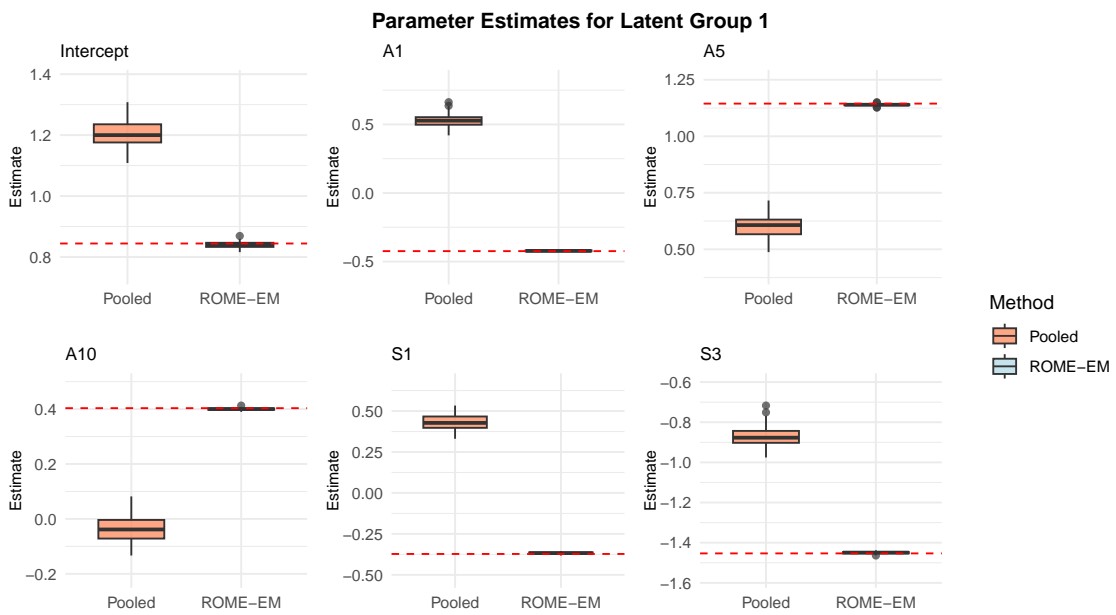

Figure 1: Box plots comparing parameter recovery by ROME-EM and pooled regression for latent group 1 (over 100 simulations). Red dashed lines denote ground truth values for each parameter.

**Results** Across 100 simulation replications, ROME-EM demonstrated superior parameter recovery and worst-group performance compared to pooled regression. Figure 1 illustrates the parameter estimation accuracy for a representative latent group, showing that ROME-EM estimates (blue) consistently cluster near the true values (red dashed lines) while pooled estimates (coral) exhibit substantial bias, particularly for group-specific parameters. The remaining results for the other three latent groups are available in Appendix A.2 which show the same patterns. The results also show ROME-EM's significant improvement in worst-group

performance. As shown in Figure 2, the worst-group MSE decreased from a mean of 36.56 under pooled regression to a mean 32.59 with ROME-EM (optimal constraint), representing a 10.58% reduction (paired t-test: $P$-value: $< 0.001$).

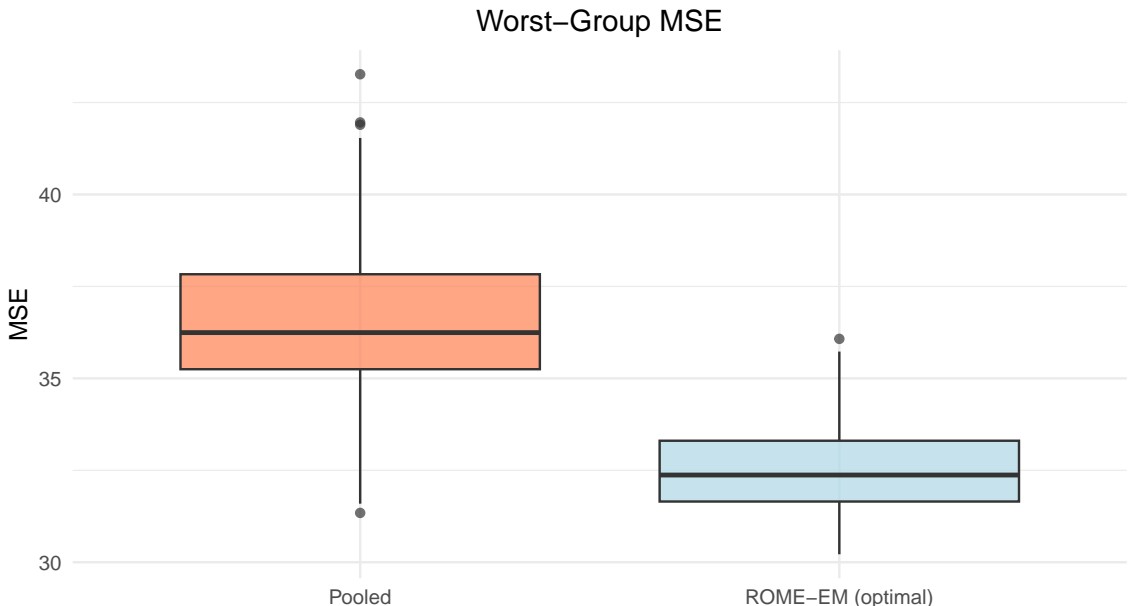

Figure 2: Box plots comparing worst-group MSE over 100 simulations.

## 4.2 REAL-DATA ANALYSIS

**Results**    We evaluated our approach on three real-world datasets from diverse domains: the Law School Admissions dataset (Wightman, 1999), the Communities & Crime dataset (Redmond, 2002), and the American Community Survey Public Use Microdata Sample (ACS PUMS) (Bureau, 2003), accessed via the Folk-tables library (Ding et al., 2021). For each dataset, we compared five models: (1) Baseline MLP: standard MLP using all features ($S$ and $A$), (2) Baseline MLP-Fair: MLP restricted to non-sensitive features ($A$) only, (3) Baseline MLP-DRO: an oracle DRO baseline that has access to discretized sensitive attributes during training. (4) Vanilla MoE: mixture-of-experts using all features for both gating and prediction, (5) ROME-MoE-S: proposed method using only $S$ for latent group gating and only $A$ for expert prediction (6) ROME-MoE-AS: proposed method using both $S$ and $A$ for latent group gating and only $A$ for expert prediction. We consider a model 'fair' only when $S$ is not directly used for outcome prediction, as this could raise ethical and legal considerations. As a result, only models (2), (3), (5) and (6) are considered being fair.

We set the DRO parameter $\alpha = 0.05$ for all experiments, balancing average performance (95% weight) with worst-group robustness (5% weight). An ablation study validates this choice (Section 4.2). We report two evaluation metrics: overall MSE on the entire test set and worst-group MSE. Since ground-truth groups are unknown in real data, we evaluate worst-group performance using intersectional subgroups. These sub-groups are defined by the cross-product of sensitive attribute categories—using natural categories for discrete variables and quartiles for continuous variables.

Tables 1 and 2 present our main results across three real-world datasets. Despite strong performance in simulations, ROME-EM encountered challenges on these particular real datasets: without clear domain

Table 1: Overall and worst-group Mean Squared Error (MSE) on test sets, averaged over 10 random seeds (mean $\pm$ standard error). Best performing fair models are shown in **bold**. Statistical significance determined using paired two-sample t-tests (n=10 seeds). For worst-group MSE, ROME methods are compared to Baseline MLP-Fair: $^{*}p < 0.05$, $^{**}p < 0.01$, $^{***}p < 0.001$. For overall MSE, ROME methods are compared to the best-performing baseline: $^{ns}$ indicates no significant difference ($p > 0.05$).

| Dataset | Model | Fair | Overall MSE | Worst-Group MSE |
|---|---|---|---|---|
| Law School Admissions Council | Baseline MLP | | $0.7196 \pm 0.0006$ | $0.8184 \pm 0.0027$ |
| | Baseline MLP - Fair | ✓ | $0.7368 \pm 0.0005$ | $0.8300 \pm 0.0012$ |
| | Baseline MLP - DRO | ✓ | $0.7357 \pm 0.0005$ | $0.7770 \pm 0.0041$ |
| | Vanilla MoE | | $0.7209 \pm 0.0007$ | $0.8233 \pm 0.0018$ |
| | ROME-MoE-S | ✓ | $0.7217 \pm 0.0014^{ns}$ | $0.8144 \pm 0.0011^{***}$ |
| | ROME-MoE-AS | ✓ | $\mathbf{0.7200 \pm 0.0015}^{ns}$ | $\mathbf{0.8127 \pm 0.0023}^{***}$ |
| Communities and Crime | Baseline MLP | | $0.0205 \pm 0.0002$ | $0.0285 \pm 0.0005$ |
| | Baseline MLP - Fair | ✓ | $0.0235 \pm 0.0002$ | $0.0326 \pm 0.0004$ |
| | Baseline MLP - DRO | ✓ | $0.0234 \pm 0.0009$ | $0.0330 \pm 0.0018$ |
| | Vanilla MoE | | $0.0207 \pm 0.0002$ | $0.0286 \pm 0.0005$ |
| | ROME-MoE-S | ✓ | $0.0207 \pm 0.0002^{ns}$ | $0.0287 \pm 0.0004^{***}$ |
| | ROME-MoE-AS | ✓ | $\mathbf{0.0204 \pm 0.0001}^{ns}$ | $\mathbf{0.0271 \pm 0.0002}^{***}$ |
| American Community Survey Public Use Microdata Sample | Baseline MLP | | $0.0044 \pm 0.0000$ | $0.0046 \pm 0.0000$ |
| | Baseline MLP - Fair | ✓ | $0.0053 \pm 0.0000$ | $0.0053 \pm 0.0000$ |
| | Baseline MLP - DRO | ✓ | $0.0052 \pm 0.0000$ | $0.0053 \pm 0.0000$ |
| | Vanilla MoE | | $0.0047 \pm 0.0000$ | $0.0048 \pm 0.0000$ |
| | ROME-MoE-S | ✓ | $\mathbf{0.0045 \pm 0.0001}^{ns}$ | $\mathbf{0.0047 \pm 0.0001}^{***}$ |
| | ROME-MoE-AS | ✓ | $0.0047 \pm 0.0001$ | $0.0049 \pm 0.0000^{***}$ |

knowledge to guide initial hard group assignments or strong mixture structure in the data, the EM algorithm either converged to degenerate solutions with identical parameters across all groups—effectively reducing to pooled regression—or failed to converge entirely. This suggests that ROME-EM may be most suitable for applications where domain expertise can inform initialization or where the underlying mixture structure is more pronounced than in our three datasets. The linear mixture model assumptions appear too restrictive for the complex heterogeneity present in these particular real-world scenarios. In contrast, ROME-MoE variants, which learn group structure end-to-end without requiring initialization, successfully improved worst-group performance across all datasets.

We present representative subgroup partitioning results for each dataset in Tables 1 and 2. Two additional partitioning schemes per dataset were evaluated as ablation studies 4.2, with full results available in Appendix B, showing consistent improvements across all schemes. Complete experimental details including preprocessing, hyperparameter selection, and evaluation protocols are provided in Appendices B and D.2.

Tables 1 and 2 show that ROME variants achieve statistically significant improvements in worst-group performance across all three datasets. For overall performance, ROME maintains parity with baselines on Law School and Crime datasets ($p < 0.01$), but shows slight degradation on ACS. The ACS dataset exhibits smaller disparity between overall and worst-group metrics compared to other datasets, indicating milder initial algorithmic bias. In such settings with limited baseline disparity, worst-case optimization naturally provides smaller gains—an expected trade-off consistent with DRO theory.

**Ablation Studies** We conducted two ablation studies to validate key design choices in ROME.

Table 2: Overall and worst-group R-squared on test sets, averaged over 10 random seeds (mean $\pm$ standard error). Best performing fair models are shown in **bold**. Statistical significance determined using paired two-sample t-tests (n=10 seeds). For worst-group R-squared, ROME methods are compared to Baseline MLP-Fair: $^{*}p < 0.05$, $^{**}p < 0.01$, $^{***}p < 0.001$. For overall R-squared, ROME methods are compared to the best-performing baseline: $^{\text{ns}}$ indicates no significant difference ($p > 0.05$).

| Dataset | Model | Fair | Overall $R^2$ | Worst-Group $R^2$ |
|---|---|---|---|---|
| Law School Admissions Council | Baseline MLP | | $0.1776 \pm 0.0007$ | $0.1225 \pm 0.0006$ |
| | Baseline MLP - Fair | ✓ | $0.1579 \pm 0.0005$ | $0.0933 \pm 0.0015$ |
| | Baseline MLP - DRO | ✓ | $0.1591 \pm 0.0006$ | $0.0835 \pm 0.0014$ |
| | Vanilla MoE | | $0.1761 \pm 0.0008$ | $0.1210 \pm 0.0012$ |
| | ROME-MoE-S | ✓ | $0.1752 \pm 0.0016^{\text{ns}}$ | $0.1148 \pm 0.0024^{***}$ |
| | ROME-MoE-AS | ✓ | $\mathbf{0.1772 \pm 0.0017^{\text{ns}}}$ | $\mathbf{0.1181 \pm 0.0021}^{***}$ |
| Communities and Crime | Baseline MLP | | $0.6149 \pm 0.0029$ | $0.5274 \pm 0.0025$ |
| | Baseline MLP - Fair | ✓ | $0.5570 \pm 0.0032$ | $0.4882 \pm 0.0062$ |
| | Baseline MLP - DRO | ✓ | $0.5588 \pm 0.0164$ | $0.4901 \pm 0.0281$ |
| | Vanilla MoE | | $0.6105 \pm 0.0033$ | $0.5281 \pm 0.0030$ |
| | ROME-MoE-S | ✓ | $0.6095 \pm 0.0032^{\text{ns}}$ | $0.5270 \pm 0.0031^{***}$ |
| | ROME-MoE-AS | ✓ | $\mathbf{0.6166 \pm 0.0024^{\text{ns}}}$ | $\mathbf{0.5356 \pm 0.0036}^{***}$ |
| American Community Survey Public Use Microdata Sample | Baseline MLP | | $0.5463 \pm 0.0035$ | $0.4907 \pm 0.0047$ |
| | Baseline MLP - Fair | ✓ | $0.4554 \pm 0.0024$ | $0.4237 \pm 0.0031$ |
| | Baseline MLP - DRO | ✓ | $0.4572 \pm 0.0019$ | $0.4248 \pm 0.0034$ |
| | Vanilla MoE | | $0.5104 \pm 0.0034$ | $0.4663 \pm 0.0036$ |
| | ROME-MoE-S | ✓ | $\mathbf{0.5299 \pm 0.0078^{\text{ns}}}$ | $\mathbf{0.4779 \pm 0.0093}^{**}$ |
| | ROME-MoE-AS | ✓ | $0.5170 \pm 0.0054$ | $0.4554 \pm 0.0053^{***}$ |

**1. Impact of DRO Parameter $\alpha$.** We conducted ablation studies varying the DRO parameter $\alpha$ to validate that the DRO objective, not merely the MoE architecture, drives performance improvements. As shown in figures 7 and 3, the worst-case improvement saturates at modest $\alpha$ values (typically $0.1-0.2$), and (3) overall MSE remains relatively stable even at $\alpha = 1.0$. These findings validate our choice of $\alpha = 0.05$ for all main experiments—it captures most worst-group benefits while maintaining near-optimal average performance. The consistency across datasets with different characteristics and both architectural variants demonstrates that the DRO objective is universally beneficial, not an artifact of specific data distributions.

**2. Impact of Evaluation Subgroups:** To ensure our improvements are not artifacts of specific subgroup definitions, we evaluated each model using multiple schemes based on different combinations of subgroup derivation using sensitive attributes. For categorical variables, we use their natural categories to partition and for continuous variables, we use either their median or quartile cuts. The details can be found in Appendix C.2, which shows the same pattern as in Table 1 and Table 2.

## 5 DISCUSSION

Our results reveal important trade-offs between interpretability and robustness in fair machine learning. ROME-EM encountered convergence challenges on real datasets, often producing solutions with similar parameters across groups. This suggests that without strong prior knowledge to guide initial hard assignments for test data, linear mixture assumptions alone may be insufficient for capturing real-world heterogeneity. ROME-MoE improved worst-group performance across all datasets, demonstrating the benefits of neural architectures for discovering meaningful latent structure in complex settings. The ROME framework ad-

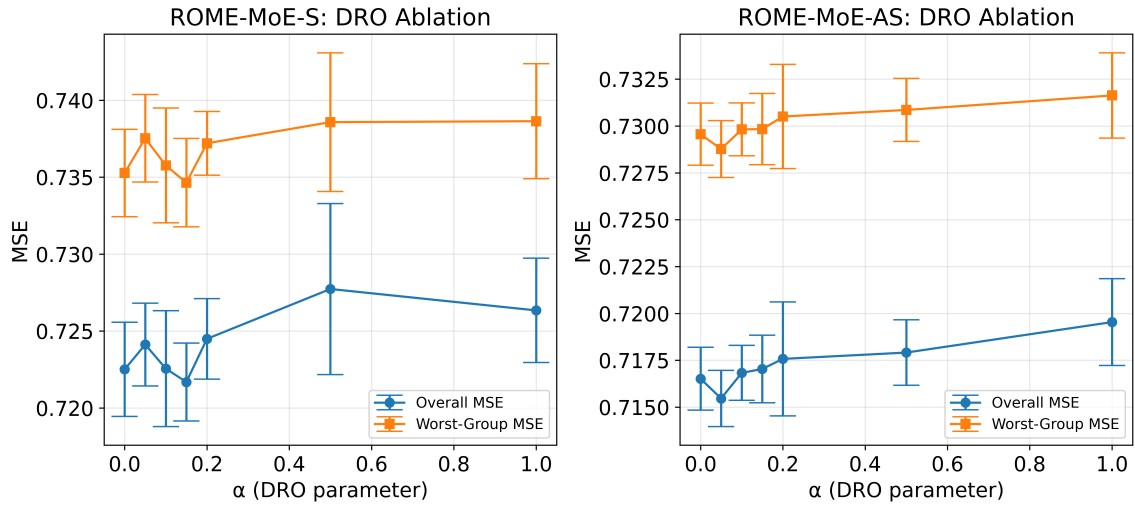

Figure 3: Effect of DRO parameter $\alpha$ on ROME-MoE performance (Law School Admissions Council dataset, 10 seeds). Similar to Crime dataset results, both variants show worst-group improvements with stable overall performance, confirming the robustness of the DRO objective across different data domains.

dresses key limitations of existing fairness approaches by discovering latent structure directly from data, naturally handling intersectionality without combinatorial explosion. By learning a small number of data-driven groups (typically 2-4), it avoids the statistical instability of auditing exponentially many subgroups while handling continuous sensitive attributes without arbitrary discretizations.

Our theoretical grounding follows the maximin aggregation framework of Wang et al. (2023), whose closed-form estimator enjoys finite-sample guarantees in the linear setting with known groups. ROME-EM applies this same aggregation rule after estimating latent components, and therefore inherits the same maximin form asymptotically under correct model specification. Extending finite-sample guarantees to the latent-group setting and to nonlinear MoE architectures is nontrivial and remains an open challenge. Establishing analogous guarantees for ROME-MoE is an important direction for future work.

One limitation of ROME is the requirement that sensitive attributes be observable during both training and validation. When such attributes are unavailable or ethically prohibited from collection, our approach cannot be directly applied. Additionally, developing principled methods for more automatic selection of the number of experts would enhance practical deployment. Despite these limitations, ROME provides a practical framework for improving worst-group performance in the common scenario where data exhibits complex latent heterogeneity involving sensitive attributes.

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

# A  SIMULATION STUDY: ADDITIONAL DETAILS

## A.1  PARAMETER SPECIFICATIONS

We follow equation 2 to generate the data, using the model parameters as detailed in Section 4.1. Specifically, we simulate features $X = (A, S) \in \mathbb{R}^p$ by drawing $X \sim \mathcal{N}(0, \Sigma_X)$. Outcomes are generated by group-specific linear models: $Y_i = \beta_{G_i}^\top [1, A_i, S_i^{\text{out}}] + \varepsilon_i$ with $\varepsilon_i \sim \mathcal{N}(0, 0.1^2)$. To model mixture shift, we optionally use distinct membership parameters $\Gamma$ (train) and $\Gamma$'(test). Finally, to mimic imperfect initialization, each true group label is flipped to a different class with small probability $r$ to form $G_0^{\text{fake}}$.

Full details and implementations are provided in script 'demo.R' of the anonymous repository.

**Group Membership Parameters ($\gamma$):** The membership parameter matrix $\gamma \in \mathbb{R}^{4\times5}$ was specified as:

$$\gamma = \begin{bmatrix} 2.0 & 2.0 & 2.0 & 2.0 & 2.0 \\ -3.0 & -2.0 & -5.0 & 0.1 & 0.1 \\ 0.1 & -10.0 & 0.1 & 0.1 & 0.1 \\ -2.0 & -2.0 & -2.0 & -2.0 & -2.0 \end{bmatrix} \tag{7}$$

**Outcome Model Coefficients ($\beta$):** The true regression coefficients were generated using a heterogeneous scaling approach. Starting with a base vector $\beta_0 = \mathbf{1}_{20}$, group-specific coefficients were created as $\beta_j = \epsilon_j \odot \beta_0$, where $\epsilon_j$ are scaling factors generated with heterogeneity parameter $\delta = 0.8$. Group intercepts were drawn uniformly from $[-1, 1]$. The final coefficient matrix $\beta^T$ (transposed for space) is:

$$\beta^T = \begin{bmatrix} 0.844 & 0.090 & 0.962 & 0.618 \\ -0.423 & 0.749 & 1.309 & 0.307 \\ 0.696 & -0.545 & 0.559 & 1.703 \\ -0.449 & 1.646 & -1.165 & 1.361 \\ -0.737 & -0.429 & 0.255 & 1.384 \\ 1.144 & -1.003 & 1.014 & -1.377 \\ 0.988 & 1.666 & -1.336 & -1.209 \\ -1.702 & -1.681 & -1.295 & 0.745 \\ 1.217 & 1.800 & -1.767 & 0.218 \\ -0.922 & -1.566 & 0.744 & -0.287 \\ 0.403 & -1.523 & 0.395 & -1.727 \\ 1.729 & 1.151 & 0.292 & 0.830 \\ -1.661 & 1.461 & 1.628 & -1.368 \\ -0.217 & 1.637 & 0.840 & -1.781 \\ -0.437 & -0.831 & -0.509 & 1.747 \\ -1.520 & -0.487 & -0.325 & -1.428 \\ -0.372 & -0.557 & 1.058 & 0.414 \\ 0.826 & -1.509 & -0.450 & 0.903 \\ -1.453 & -0.848 & -0.748 & 1.429 \\ -0.773 & 0.996 & 0.765 & 1.460 \\ -1.134 & -1.441 & 0.509 & -1.790 \end{bmatrix} \tag{8}$$

where rows 1-21 correspond to the intercept, $A_1$-$A_{15}$, and $S_1$-$S_5$ respectively, and columns 1-4 correspond to latent groups 1-4.

**Algorithm Hyperparameters:**

- EM convergence tolerance: $\tau_1 = 10^{-3}$ (parameter change), $\tau_2 = 5 \times 10^{-3}$ (line search)

- Maximum EM iterations: 100

- DRO constraint values: $\mathcal{C} \in \{1.0, 0.6, 0.5, 0.49, 0.48, 0.47, \ldots, 0.04, 0.03, 0.02\}$ (27 values total)

- Misspecification rate: 50% (proportion of observations with incorrect initial group assignment)

**Data Structure:** All features were generated from independent standard normal distributions ($\Sigma_X = \boldsymbol{I}_{20}$). The first 15 features served as non-sensitive features ($\boldsymbol{A}$), while the remaining 5 were treated as sensitive attributes ($\boldsymbol{S}$). All sensitive attributes were included in both membership ($\mathcal{I}_{\text{mem}} = \{1, 2, 3, 4, 5\}$) and outcome ($\mathcal{I}_{\text{out}} = \{1, 2, 3, 4, 5\}$) models during training, without loss of generalizibility.

## A.2 ADDITIONAL SIMULATION RESULTS

Figures 4, 5, and 6 present the additional simulation results as complementary to Figure 1 in the main text.

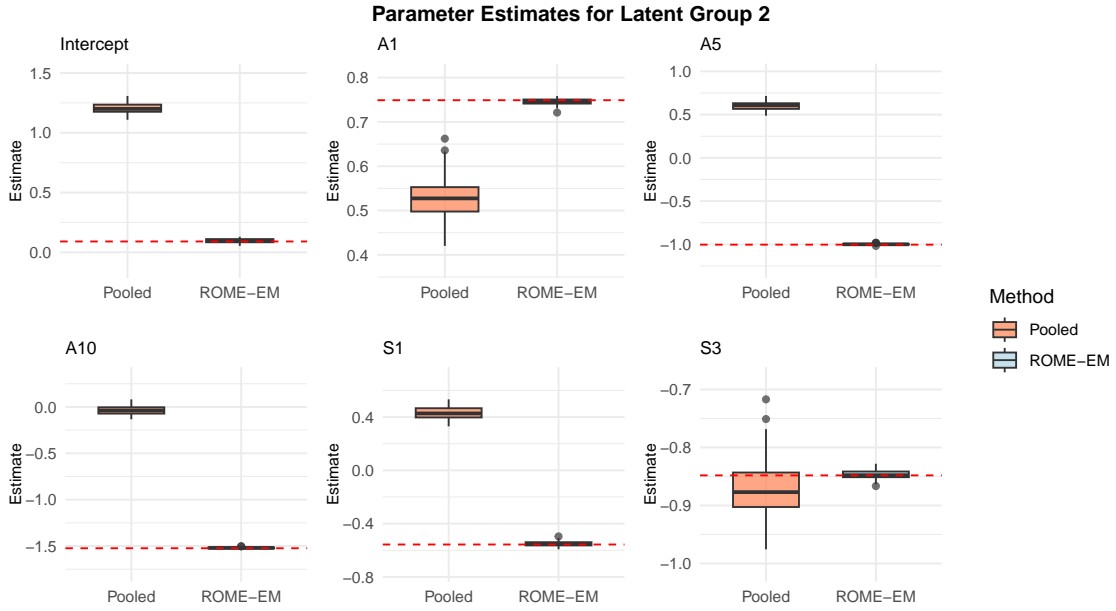

Figure 4: Box plots comparing parameter recovery by ROME-EM and pooled regression for latent group 2 (over 100 simulations). Red dashed lines denote ground truth values for each parameter.

## B DETAILS OF REAL-DATASETS

**Law School Admissions Council (Wightman, 1999)** We use 'zfygpa' as $Y$, features 'race1_black' ($S_1$), 'gender_male' ($S_2$) and 'age' ($S_3$) as $\boldsymbol{S}$; 'lsat','ugpa','fam_inc', 'tier' and 'fulltime' as $\boldsymbol{A}$.

**Communities and Crime (Redmond, 2002)** We use 'ViolentCrimesPerPop' as $Y$, features 'racepctblack' ($S_1$), 'racePctHisp' ($S_2$) and 'agePct12t21' ($S_3$) as $\boldsymbol{S}$; and features 'PctUnemployed', 'medIncome', 'Pop-Dens', 'PolicPerPop', 'MedRent', 'PctFam2Par','PctIlleg', 'LandArea' and 'pctWWage' as $\boldsymbol{A}$.

**American Community Survey Public Use Microdata Sample (Bureau, 2003)** We use 'PINCP' as $Y$, features 'SEX' ($S_1$), 'RAC1P'($S_2$), 'AGEP' ($S_3$) and 'SCHL' ($S_4$) as $\boldsymbol{S}$; and features 'COW', 'MAR', 'OCCP', 'POBP', 'RELP', 'WKHP' as $\boldsymbol{A}$.

## C ABLATION STUDIES

### C.1 IMPACT OF DRO PARAMETER

Figures 7 and 8 present the additional results for $\alpha$ ablation studies as complementary to Figure 3 in the main text.

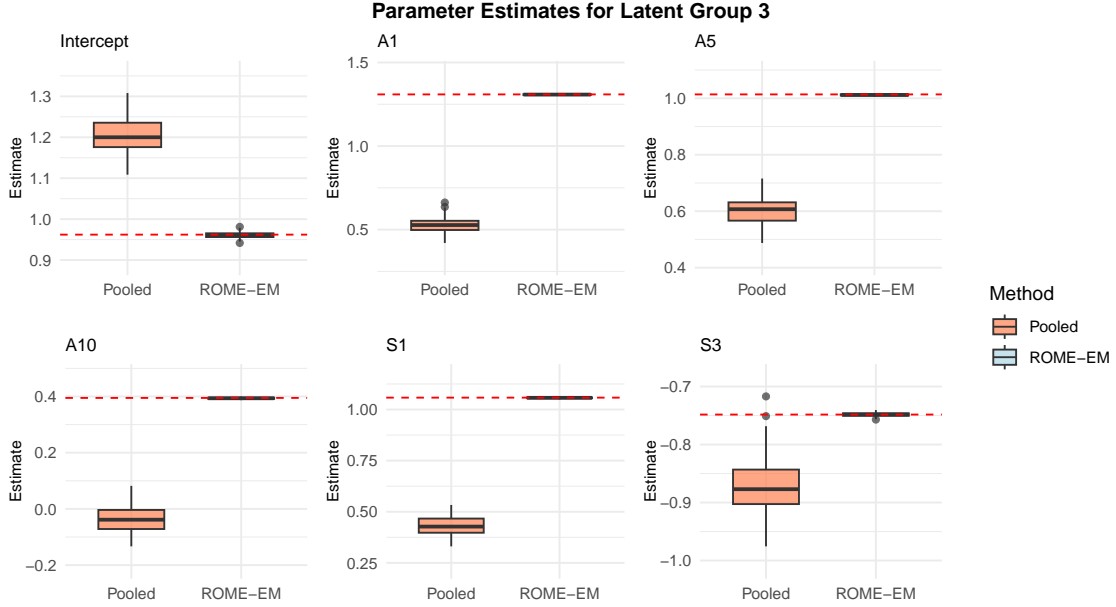

Figure 5: Box plots comparing parameter recovery by ROME-EM and pooled regression for latent group 3 (over 100 simulations). Red dashed lines denote ground truth values for each parameter.

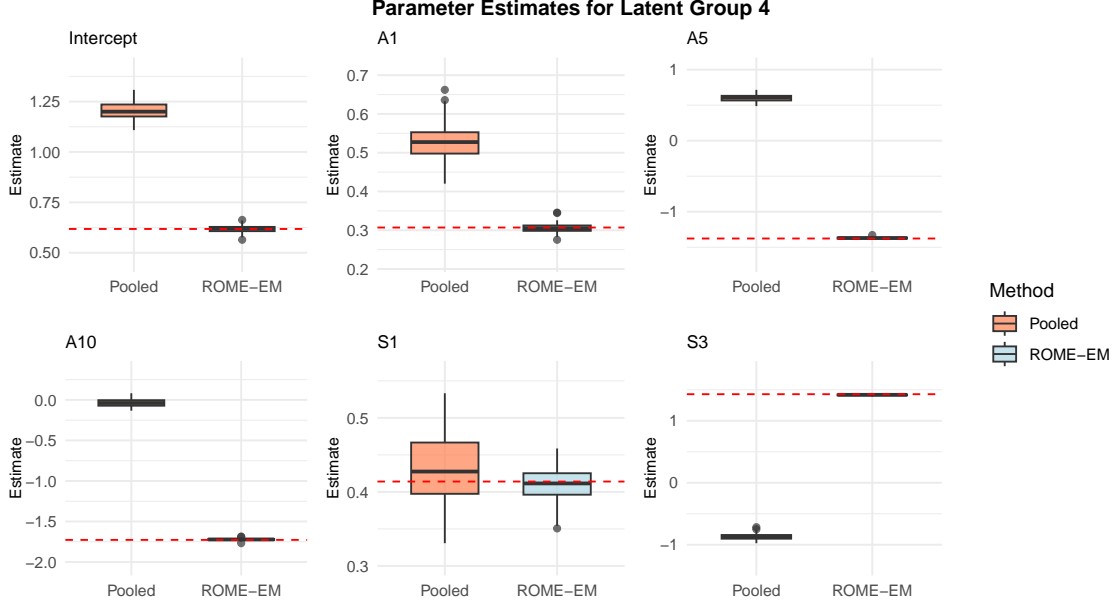

Figure 6: Box plots comparing parameter recovery by ROME-EM and pooled regression for latent group 4 (over 100 simulations). Red dashed lines denote ground truth values for each parameter.

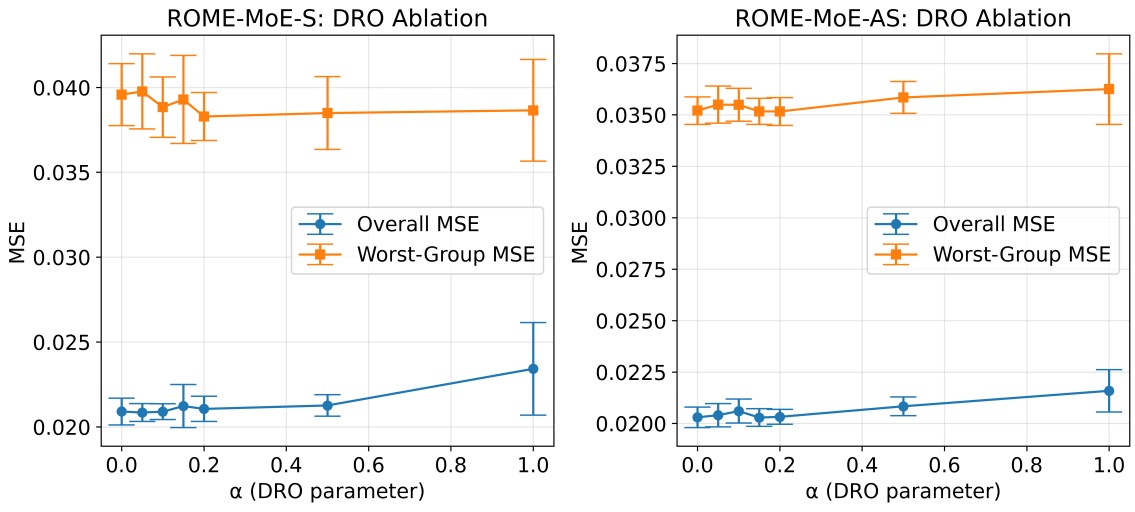

Figure 7: Effect of DRO parameter $\alpha$ on ROME-MoE performance (Communities & Crime dataset, 10 seeds). Both variants show worst-group improvements with minimal overall performance degradation as $\alpha$ increases from 0 (standard training) to 1 (pure worst-group optimization).

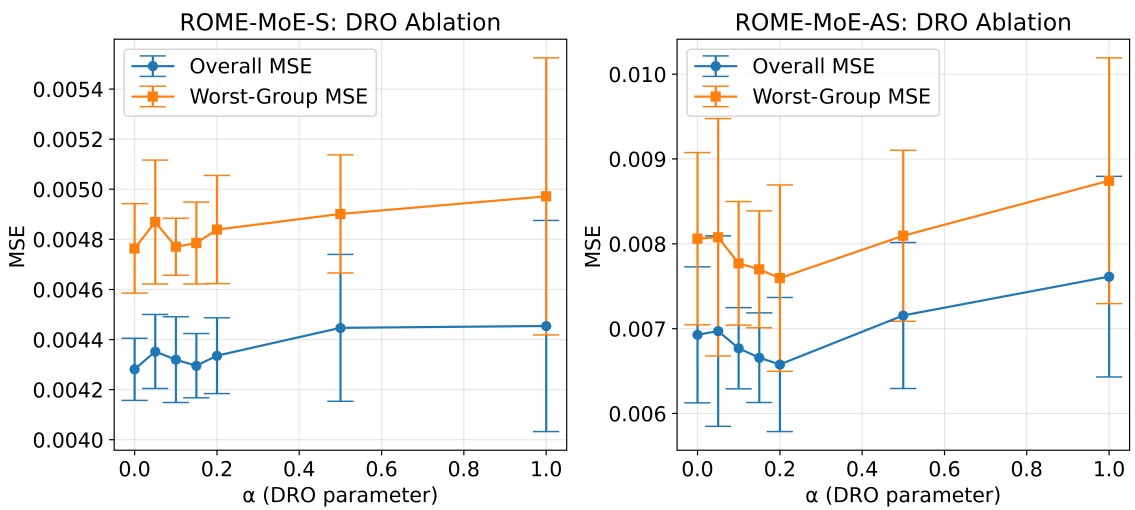

Figure 8: Effect of DRO parameter $\alpha$ on ROME-MoE performance (American Community Survey dataset, 10 seeds). Both variants show worst-group improvements with minimal overall performance degradation as $\alpha$ increases from 0 (standard training) to 1 (pure worst-group optimization).

## C.2 IMPACT OF EVALUATION SUBGROUPS

Evaluation subgroups are formed by partitioning sensitive attributes: categorical variables retain their discrete values, while continuous variables are discretized using median or quartile splits. Besides Tables 1 and 2 in main text, where we use quartile cuts for $(S_2, S_3, \text{quartile})$ for law dataset, $(S_2, S_3, \text{median})$ for

Table 3: Ablation study on evaluation subgroup schemes for Law School dataset. Results show worst-group performance across different sensitive attribute combinations, averaged over 10 random seeds (mean $\pm$ standard error). Best fair models in **bold**. Statistical significance (paired t-test, n=10): $^*p < 0.05$, $^{**}p < 0.01$, $^{***}p < 0.001$ comparing ROME variants to Baseline MLP-Fair for worst-group metrics.

| Evaluation Subgroups | Model | Fair | Worst-Group MSE | Worst-Group $R^2$ |
|---|---|---|---|---|
| $S_1, S_3$, median | Baseline MLP | | $0.7640 \pm 0.0007$ | $0.1729 \pm 0.0656$ |
| | Baseline MLP - Fair | ✓ | $0.7795 \pm 0.0047$ | $0.0402 \pm 0.0827$ |
| | Baseline MLP - DRO | ✓ | $0.7714 \pm 0.0015$ | $-0.0189 \pm 0.0080$ |
| | Vanilla MoE | | $0.7656 \pm 0.0010$ | $0.1719 \pm 0.0663$ |
| | ROME-MoE-S | ✓ | $0.7628 \pm 0.0010^{**}$ | $\mathbf{0.1710 \pm 0.0664}^{***}$ |
| | ROME-MoE-AS | ✓ | $\mathbf{0.7622 \pm 0.0015}^{***}$ | $0.1581 \pm 0.0689^{**}$ |
| $S_2, S_3$, median | Baseline MLP | | $0.7674 \pm 0.0010$ | $0.1276 \pm 0.0006$ |
| | Baseline MLP - Fair | ✓ | $0.7825 \pm 0.0015$ | $0.1029 \pm 0.0011$ |
| | Baseline MLP - DRO | ✓ | $0.7805 \pm 0.0018$ | $0.1030 \pm 0.0006$ |
| | Vanilla MoE | | $0.7681 \pm 0.0015$ | $0.1277 \pm 0.0006$ |
| | ROME-MoE-S | ✓ | $0.7675 \pm 0.0019^{***}$ | $0.1210 \pm 0.0020^{***}$ |
| | ROME-MoE-AS | ✓ | $\mathbf{0.7644 \pm 0.0016}^{***}$ | $\mathbf{0.1235 \pm 0.0012}^{***}$ |

Table 4: Ablation study on evaluation subgroup schemes for Communities & Crime dataset. Results show worst-group performance across different sensitive attribute combinations, averaged over 10 random seeds (mean $\pm$ standard error). Best fair models in **bold**. Statistical significance (paired t-test, n=10): $^*p < 0.05$, $^{**}p < 0.01$, $^{***}p < 0.001$ comparing ROME variants to Baseline MLP-Fair for worst-group metrics.

| Evaluation Subgroups | Model | Fair | Worst-Group MSE | Worst-Group $R^2$ |
|---|---|---|---|---|
| $S_1, S_2$, median | Baseline MLP | | $0.0402 \pm 0.0005$ | $0.2106 \pm 0.0081$ |
| | Baseline MLP - Fair | ✓ | $0.0418 \pm 0.0003$ | $-0.0542 \pm 0.0425$ |
| | Baseline MLP - DRO | ✓ | $0.0423 \pm 0.0017$ | $-0.0292 \pm 0.1333$ |
| | Vanilla MoE | | $0.0404 \pm 0.0007$ | $0.2100 \pm 0.0084$ |
| | ROME-MoE-S | ✓ | $0.0407 \pm 0.0005$ | $\mathbf{0.2005 \pm 0.0061}^{***}$ |
| | ROME-MoE-AS | ✓ | $\mathbf{0.0369 \pm 0.0002}^{***}$ | $0.1406 \pm 0.0108^{**}$ |
| $S_1, S_3$, median | Baseline MLP | | $0.0332 \pm 0.0004$ | $0.1847 \pm 0.0076$ |
| | Baseline MLP - Fair | ✓ | $0.0385 \pm 0.0004$ | $0.1352 \pm 0.0156$ |
| | Baseline MLP - DRO | ✓ | $0.0389 \pm 0.0015$ | $0.1500 \pm 0.0466$ |
| | Vanilla MoE | | $0.0339 \pm 0.0007$ | $0.1885 \pm 0.0079$ |
| | ROME-MoE-S | ✓ | $0.0339 \pm 0.0004^{***}$ | $\mathbf{0.1796 \pm 0.0051}^{**}$ |
| | ROME-MoE-AS | ✓ | $\mathbf{0.0319 \pm 0.0002}^{***}$ | $0.1520 \pm 0.0050$ |

crime dataset and $S_2$ for American Community Survey dataset, where feature details are available in B, we conduct ablation studies by further using different evaluation subgroups.

# D NEURAL NETWORK DETAILS

## D.1 NEURAL NETWORK ARCHITECTURE

All MoE-based models (Vanilla MoE, ROME-MoE-S, ROME-MoE-AS) use standard two-layer multi-layer perceptrons (MLPs) for both the experts and the gating network. Each expert has the structure Linear{ReLU{Linear and outputs a scalar prediction. The Vanilla MoE experts take $(A, S)$ as input,

Table 5: Ablation study on evaluation subgroup schemes for American Community Survey Public Use Microdata Sample dataset. Results show worst-group performance across different sensitive attribute combinations, averaged over 10 random seeds (mean $\pm$ standard error). Best fair models in **bold**. Statistical significance (paired t-test, n=10): $^*p < 0.05$, $^{**}p < 0.01$, $^{***}p < 0.001$ comparing ROME variants to Baseline MLP-Fair for worst-group metrics.

| Evaluation Subgroups | Model | Fair | Worst-Group MSE | Worst-Group $R^2$ |
|---|---|---|---|---|
| $S_1$ | Baseline MLP | | $0.0047 \pm 0.0000$ | $0.4870 \pm 0.0037$ |
| | Baseline MLP - Fair | ✓ | $0.0055 \pm 0.0001$ | $0.4026 \pm 0.0054$ |
| | Baseline MLP - DRO | ✓ | $0.0056 \pm 0.0000$ | $0.4001 \pm 0.0035$ |
| | Vanilla MoE | | $0.0052 \pm 0.0000$ | $0.4451 \pm 0.0038$ |
| | ROME-MoE-S | ✓ | $\mathbf{0.0049 \pm 0.0001}^{***}$ | $\mathbf{0.4678 \pm 0.0060}^{***}$ |
| | ROME-MoE-AS | ✓ | $0.0051 \pm 0.0001^{***}$ | $0.4499 \pm 0.0065^{***}$ |
| $S_1$ and $S_2$ | Baseline MLP | | $0.0049 \pm 0.0000$ | $0.4445 \pm 0.0053$ |
| | Baseline MLP - Fair | ✓ | $0.0056 \pm 0.0001$ | $0.3764 \pm 0.0067$ |
| | Baseline MLP - DRO | ✓ | $0.0057 \pm 0.0000$ | $0.3821 \pm 0.0061$ |
| | Vanilla MoE | | $0.0052 \pm 0.0000$ | $0.4039 \pm 0.0049$ |
| | ROME-MoE-S | ✓ | $\mathbf{0.0050 \pm 0.0001}^{***}$ | $\mathbf{0.4328 \pm 0.0080}^{**}$ |
| | ROME-MoE-AS | ✓ | $0.0052 \pm 0.0001^{***}$ | $0.4074 \pm 0.0080^{*}$ |

while the fair variants (ROME-MoE-S / ROME-MoE-AS) use only $A$ as expert input. The gating network is also a two-layer MLP, taking either $S$ (ROME-MoE-S) or $(A, S)$ (ROME-MoE-AS) as input and producing $G$ logits, followed by a softmax to obtain mixture weights. The final prediction is the weighted combination of expert outputs. Hyperparameter search spaces and selected values (e.g., hidden sizes $\{32, 64, 128\}$, number of experts $G \in \{2, 3, 4\}$) are provided in Tables 6, 7 and 8.

All baseline MLP models use a standard two-layer multilayer perceptron (MLP) of the form Linear $\rightarrow$ ReLU $\rightarrow$ Linear, producing a scalar prediction. The variants differ only in their input features and training objective: Baseline MLP–Full (unfair): experts receive the full feature vector (A, S); Baseline MLP–Fair: experts receive only A, matching the fairness constraint used in ROME-MoE; Baseline MLP–DRO: same architecture as Baseline MLP–Fair, but trained with a DRO loss combining average and worst-group MSE, where groups are defined using (discretized) sensitive attributes S. Hyperparameter search spaces (hidden sizes $\{32, 64, 128\}$, learning rates, etc.) are identical to those used for MoE and are summarized in Tables 6–8.

## D.2 HYPERPARAMETER TUNING

For each dataset, we performed a grid search to select the best hyperparameters for all models. The search space and the final selected values for each dataset are detailed in Tables 6, 7 and 8.

## E LLM USAGE DECLARATION

We used Claude (Anthropic) as a writing-assistance tool to improve grammar and clarity during manuscript preparation. All research ideas, designs, and analyses were conducted by the authors, who take full responsibility for the accuracy and integrity of the content.

Table 6: Hyperparameter tuning results for Law School Admissions Council

| Hyperparameter | Search Space | Best Value |
|---|---|---|
| *BaselineMLP* | | |
| Learning Rate | {1e-2, 1e-3, 1e-4} | 1e-4 |
| Hidden Size | {32, 64, 128} | 64 |
| *BaselineMLP-Fair* | | |
| Learning Rate | {1e-2, 1e-3, 1e-4} | 1e-3 |
| Hidden Size | {32, 64, 128} | 32 |
| *BaselineMLP-DRO* | | |
| Learning Rate | {1e-2, 1e-3, 1e-4} | 1e-3 |
| Hidden Size | {32, 64, 128} | 32 |
| *Vanilla MoE* | | |
| Learning Rate | {1e-2, 1e-3, 1e-4} | 1e-4 |
| Hidden Size (expert) | {32, 64, 128} | 32 |
| Hidden Size (gating) | {32, 64, 128} | 64 |
| Number of Experts | {2, 3, 4} | 4 |
| *ROME-MoE-S* | | |
| Learning Rate | {1e-2, 1e-3, 1e-4} | 1e-4 |
| Hidden Size (expert) | {32, 64, 128} | 128 |
| Hidden Size (gating) | {32, 64, 128} | 16 |
| Number of Experts | {2, 3, 4} | 2 |
| *ROME-MoE-AS* | | |
| Learning Rate | {1e-2, 1e-3, 1e-4} | 1e-4 |
| Hidden Size (expert) | {32, 64, 128} | 64 |
| Hidden Size (gating) | {32, 64, 128} | 64 |
| Number of Experts | {2, 3, 4} | 2 |

Table 7: Hyperparameter tuning results for Communities and Crime dataset

| Hyperparameter | Search Space | Best Value |
|---|---|---|
| *BaselineMLP* | | |
| Learning Rate | {1e-2, 1e-3, 1e-4} | 1e-2 |
| Hidden Size | {32, 64, 128} | 128 |
| *BaselineMLP-Fair* | | |
| Learning Rate | {1e-2, 1e-3, 1e-4} | 1e-2 |
| Hidden Size | {32, 64, 128} | 64 |
| *BaselineMLP-DRO* | | |
| Learning Rate | {1e-2, 1e-3, 1e-4} | 1e-2 |
| Hidden Size | {32, 64, 128} | 64 |
| *Vanilla MoE* | | |
| Learning Rate | {1e-2, 1e-3, 1e-4} | 1e-2 |
| Hidden Size (expert) | {32, 64, 128} | 64 |
| Hidden Size (gating) | {32, 64, 128} | 32 |
| Number of Experts | {2, 3, 4} | 2 |
| *ROME-MoE-S* | | |
| Learning Rate | {1e-2, 1e-3, 1e-4} | 1e-2 |
| Hidden Size (expert) | {32, 64, 128} | 64 |
| Hidden Size (gating) | {32, 64, 128} | 16 |
| Number of Experts | {2, 3, 4} | 3 |
| *ROME-MoE-AS* | | |
| Learning Rate | {1e-2, 1e-3, 1e-4} | 1e-3 |
| Hidden Size (expert) | {32, 64, 128} | 32 |
| Hidden Size (gating) | {32, 64, 128} | 16 |
| Number of Experts | {2, 3, 4} | 2 |

Table 8: Hyperparameter tuning results for American Community Survey Public Use Microdata Sample dataset

| Hyperparameter | Search Space | Best Value |
|---|---|---|
| *BaselineMLP* | | |
| Learning Rate | {1e-2, 1e-3, 1e-4} | 1e-3 |
| Hidden Size | {32, 64, 128} | 64 |
| *BaselineMLP-Fair* | | |
| Learning Rate | {1e-2, 1e-3, 1e-4} | 1e-3 |
| Hidden Size | {32, 64, 128} | 32 |
| *BaselineMLP-DRO* | | |
| Learning Rate | {1e-2, 1e-3, 1e-4} | 1e-3 |
| Hidden Size | {32, 64, 128} | 32 |
| *Vanilla MoE* | | |
| Learning Rate | {1e-2, 1e-3, 1e-4} | 1e-4 |
| Hidden Size (expert) | {32, 64, 128} | 128 |
| Hidden Size (gating) | {32, 64, 128} | 16 |
| Number of Experts | {2, 3, 4} | 4 |
| *ROME-MoE-S* | | |
| Learning Rate | {1e-2, 1e-3, 1e-4} | 1e-2 |
| Hidden Size (expert) | {32, 64, 128} | 128 |
| Hidden Size (gating) | {32, 64, 128} | 32 |
| Number of Experts | {2, 3, 4} | 2 |
| *ROME-MoE-AS* | | |
| Learning Rate | {1e-2, 1e-3, 1e-4} | 1e-4 |
| Hidden Size (expert) | {32, 64, 128} | 128 |
| Hidden Size (gating) | {32, 64, 128} | 32 |
| Number of Experts | {2, 3, 4} | 4 |

