# OpenReview forum: "Robust Mixture Models for Algorithmic Fairness Under Latent Heterogeneity"
_ICLR.cc/2026/Conference — Submitted to ICLR 2026_

### Official Review · Reviewer_Y6xa · 2025-10-30

**Soundness:** 2
**Presentation:** 2
**Contribution:** 2
**Rating:** 4
**Confidence:** 3

**Summary:**

The paper introduces ROME (Robust Mixture Ensemble), a novel framework that addresses algorithmic fairness by discovering and optimizing for *latent* subgroups, rather than relying on predefined demographic categories. By integrating Distributionally Robust Optimization (DRO), the method explicitly optimizes for worst-group performance, providing a more robust and practical approach to equity in machine learning models.

**Strengths:**

1. Dual-Model Framework: The paper provides two flexible implementations: ROME-EM for linear models and ROME-MoE for non-linear, neural network settings.
2. Closed-Form Solution: The ROME-EM approach offers a closed-form solution for the optimal robust predictor in linear contexts.
3. Non-Linear Adaptation: The ROME-MoE model effectively adapts the fairness framework to complex, non-linear problems using a Mixture-of-Experts architecture.
4. Comprehensive Validation: The methodology is validated using both synthetic data (confirming its ability to recover true group parameters) and three real-world datasets (demonstrating practical utility).

**Weaknesses:**

Although the paper is interesting and its contributions are relevant, it has several weak points that should be addressed prior to publication. Specifically:

1. Poor Presentation: The paper's notation is confusing, particularly in Section 2.3. For example, $\mathcal{C}$ is introduced but never used, the exact definition of $Q_X$ is unclear, $\mathbb{P}$ is never introduced, the definition of $\mathcal{H}$ is repeated, and the distinction between $\hat{f}$, $f^*$, and $f^{(k)}$ is misleading. This is compounded by other symbols that are either not introduced or introduced but never used. Another example, outside this section, is the definition of $\mathbf{X}$ and $X_i$: the former is never introduced, and the latter seems to be a vector (features of sample $i$) rather than a matrix. The overall flow and clarity of the paper should be thoroughly revised.
2. Minor Contribution of ROME-MoE: The differences between Vanilla-MoE and ROME-MoE appear to lie just in the loss function's use of robust optimization (accounting for worst-case performance). While this is a contribution, it is a minor one, and its relationship to the linear case is also tenuous. It would be interesting to see an intermediate model where expert weights are learned via some version of the linear model, while still using neural networks for the experts to capture non-linearities.
3. Overstated "Guarantees": The authors mention “theoretical guarantees” and “convergence guarantees” for the linear model in Section 3, but these are not provided. A closed-form solution for one part of the algorithm is not a theoretical guarantee, and convergence is never proven. The authors should soften these claims regarding the linear model.
4. Lack of Clarity (Linear Model): It is unclear why hard prior group assignments are needed, only to then compute a “membership probability” (is this similar to a soft assignment?). It is also unclear if these probabilities are computed using Equation (1), as the notation changes (introducing $\gamma$). Furthermore, the group assignments $z_i$ mentioned in the initialization should be $z_{ij}$ to be consistent with the notation in Section 2.1.
5. Limited SOTA Comparisons: The comparisons are limited to Vanilla-MoE (where differences are minor, as discussed) and standard MLPs. The authors should strengthen the numerical comparisons against other SOTA methods, even if they are not optimized for fairness or do not use group-specific architectures.
6. Weak Experimental Design: The authors state that “ground-truth groups are unknown in real data,” but datasets exist where sensitive attributes (e.g., race) provide clear group divisions. Using such data would offer a valuable benchmark, as the number of groups and assignments would be known (likely removing the need for a gating mechanism) and would better isolate the performance gains from the robust optimization. Furthermore, the linear model's performance is not evaluated on any real-world data. If space is a concern, the figures displaying parameter estimates are unnecessary when MSE is already reported.
7. Missing Fairness Metrics: While the structural fairness approach is valid, the paper would benefit from comparing its predictions against baselines using established fairness metrics from the literature.
8. Unclear Synthetic Setup: For the synthetic experiment, "misspecified" starting conditions are created by setting 50% of observations to incorrect groups. It is unclear if this refers to the initial group assignments for parameter estimation. If so, it's not obvious why this method was chosen over a standard uniform random assignment.
9 Robustness to G: The paper mentions that the number of groups (G) can be fine-tuned using information criteria, but it never analyzes the algorithm's robustness to an incorrect estimation of G.

Minor Issues:

* In the E-step, the term "posterior responsibilities" is used. Is this standard, or should it be "posterior probabilities"?
* In Algorithm 1, if the gating network $g$ already outputs a probability distribution, the `Softmax` call in lines 9 and 11 appears redundant. Also, in line 13, estimates for groups whose coefficient is not significant are not needed, meaning a computational overhead for the algorithm.
* The authors mention generating “covariates” in the synthetic experiment, and it is unclear what this refers to.
* In Table 2’s caption, "MSE" is mentioned, while the table reports $R^2$ values.

**Questions:**

See weaknesses section.

---

> ### Author Response · Authors · 2025-11-20
>
> Thank you for the thoughtful and constructive feedback. We have carefully revised the manuscript to address each point. Below we provide a point-by-point response. All proposed revisions have been incorporated into the revised manuscript (major changes highlighted in blue and minor edits reflected directly).
>
>
> ## Weakness
>
> **W1. Poor Presentation.**
> **Response:** Thank you for pointing this out. We have corrected the notation issues in Section 2 and improved the wording to enhance flow and clarity. These changes are reflected in the revised PDF.
>
> **W2. Minor Contribution of ROME-MoE.**
> **Response:** Thank you for this insightful comment. Our goal in ROME-MoE is to extend the robust maximin aggregation principle—well-motivated in the linear setting—to Mixture-of-Experts models for latent-group fairness under subgroup shift, a context in which MoE architectures have been scarcely explored. Although the architectural difference from Vanilla-MoE appears small, the robust objective fundamentally alters the learned mixture weights and consistently improves worst-group performance in our experiments.
> We agree with you that intermediate hybrid models (e.g., partially linear or EM-inspired gating combined with neural experts) represent an interesting direction. There also exists work that incorporates EM-style components into neural networks in survival analysis[1]; while it does not address fairness or subgroup robustness, it supports your suggestion that such hybrids are a promising research avenue. Due to space and time constraints, we leave these extensions for future work and now explicitly mention this direction in the revised manuscript.
> More broadly, our contribution lies not in proposing an entirely new architecture, but in providing a principled and theoretically grounded extension of robust aggregation to MoE models in a fairness setting, together with empirical evidence demonstrating that this materially improves worst-group performance. We have clarified this motivation in the revision to better convey the novelty and significance of ROME-MoE.
>
> [1] Nagpal, C., Yadlowsky, S., Rostamzadeh, N., & Heller, K. (2021, October). Deep cox mixtures for survival regression. In Machine Learning for Healthcare Conference (pp. 674-708). PMLR.
>
> **W3. Overstated "Guarantees".**
> **Response:** Thank you for raising this point. We agree that the original wording was overly strong. We have softened the language and clarified that our main theoretical component is the closed-form robust aggregation step, rather than a global convergence theorem for EM. Please see the revised manuscript (Section 3, marked in blue).
>
>
> **W4. Lack of Clarity (Linear Model).**
> **Response:** Thank you for pointing this out. In the linear ROME-EM model, the initial hard group assignments serve only as a warm start for the EM procedure. The membership probabilities referred to in Section 3 are the standard EM responsibilities computed using Equation (1). We have clarified this connection in the revised text.
> We also agree that the notation was inconsistent, and we have revised Section 2.1 and Section 3 to use $z_{ij}$ consistently throughout. (Since these are minor typographical corrections, we did not highlight them in blue.) These edits should make the relationship between the hard initialization and the soft EM responsibilities clearer.
>
>
> **W5. Limited SOTA Comparisons**
> **Response:** Thank you for this suggestion. We agree that including additional baselines strengthens the empirical comparison. In the revised manuscript, we have added an oracle-group baseline based on GroupDRO applied to the same MLP architecture as our existing baselines. Here, groups are defined using discretized or intersectional sensitive attributes (e.g., race × gender), which enables us to incorporate a widely used fairness-oriented DRO method even though it assumes observed groups. This provides a more comprehensive comparison and helps isolate the contribution of the robust aggregation mechanism in ROME-MoE.
> The details of this new baseline have been added to the Experiment section, and the new results are reported in Tables 2–5 of the revised manuscript. As shown, this baseline is still outperformed by our proposed methods, highlighting the advantages of ROME-MoE.

---

> > ### Author Response · Authors · 2025-11-20
> >
> > **W6. Weak Experimental Design.**
> > **Response:** Thank you for these helpful comments.
> > 1) We clarify that our real-world datasets (e.g., ACS, MIMIC) already contain sensitive attributes such as race and gender, and our evaluation (worst-group MSE and group disparity metrics) is computed over these subpopulations. To address your suggestion directly, we now include an oracle-group baseline (MLP + GroupDRO) that makes use of these known sensitive groups. This provides a benchmark where group memberships are observed and helps isolate the gains attributable specifically to the robust aggregation step used in ROME-MoE.
> > 2) Regarding the linear ROME-EM model, we intentionally evaluate it primarily on synthetic data where the true parameters are known. This setting allows us to study parameter recovery and behavior under misspecification—properties that cannot be meaningfully assessed on high-dimensional real data. We also emphasize that this work considers both prediction-focused tasks and non-prediction tasks (e.g., statistical inference, causal analysis) as equally important, and parameter-estimation results provide insights into inferential stability under latent heterogeneity. Since ROME-MoE is the model intended for complex real-world prediction scenarios, we focus real-data experiments on ROME-MoE and retain the parameter-estimation figures as they capture complementary inferential behavior not reflected by predictive MSE alone. These clarifications have been added to the revised manuscript.
> >
> > We hope that the additional experiments added in response to W5 and W9 further reinforce the empirical evaluation and address the reviewer’s concerns regarding experimental design.
> >
> >
> > **W7. Missing Fairness Metrics.**
> > **Response:** Thank you for the suggestion. Most “established” fairness metrics in the literature—such as demographic parity, equalized odds, predictive parity, or FPR/FNR gaps—are defined for binary classification, and even within that setting different papers adopt different metrics[1,2,3]. These definitions do not extend naturally to regression, and applying them directly to continuous outcomes can be ill-defined or misleading.
> > In regression settings, the standard fairness criteria used in subgroup-robust learning and GroupDRO are group-wise risk measures (e.g., worst-group MSE, group gaps, risk disparities). These are exactly the metrics we report in our real-data experiments.
> > We have added additional details in Section 2.1 of the revised manuscript to clarify these points for the readers.
> > In addtion, to strengthen the comparison, we additionally include an oracle-group baseline (MLP + GroupDRO) using known sensitive groups (see responses to W5 and W6), providing a clearer benchmark for evaluating group disparities in the regression setting.
> >
> > [1] Bird, S., Dudík, M., Edgar, R., Horn, B., Lutz, R., Milan, V., ... & Walker, K. (2020). Fairlearn: A toolkit for assessing and improving fairness in AI
> >
> > [2] Hong, C., Liu, M., Wojdyla, D. M., Hickey, J., Pencina, M., & Henao, R. (2024). Trans-balance: Reducing demographic disparity for prediction models in the presence of class imbalance. Journal of biomedical informatics, 149, 104532.
> >
> > [3] Du, W., Xu, D., Wu, X., & Tong, H. (2021). Fairness-aware agnostic federated learning. In Proceedings of the 2021 SIAM International Conference on Data Mining (SDM) (pp. 181-189). Society for Industrial and Applied Mathematics.

---

> > > ### Author Response · Authors · 2025-11-20
> > >
> > > **W8. Unclear Synthetic Setup.**
> > > **Response:** Thank you for pointing this out. Yes, the “50% incorrect” assignment refers to the initial hard group labels used to warm-start ROME-EM. We chose this setup deliberately to create a controlled level of misspecification rather than a fully uniform random initialization. A uniform random assignment would typically assign nearly all observations to incorrect groups, making the initial responsibilities almost uninformative and leading to highly unstable EM updates. In contrast, flipping approximately 50% of labels provides a moderately misspecified but still structured starting point that allows us to examine the robustness of ROME-EM to imperfect initialization.
> > > We have clarified this rationale and the precise initialization scheme in the revised manuscript.
> > >
> > >
> > > **W9. Robustness to $G$.**
> > > **Response:**  Thank you for your comment. We respectfully note that G in our framework is a hyperparameter, and the question of 'robustness to incorrect G' is essentially a model misspecification question, which has been extensively studied in the mixture model literature [1,2].
> > > For our neural network-based ROME-MoE approach, we conducted sensitivity analysis across $G \in {2, 3, 4}$ using Crime data and found that: (1) The fine-tuned value of G indeed achieves the best worst-group performance compared to the two other candidates (2%-10% better), confirming that standard hyperparameter selection via cross-validation can effectively identify optimal G. (2) Performance degrades gracefully with suboptimal G, similar to using suboptimal hyperparameters in standard neural networks.
> > > For ROME-EM, model misspecification with incorrect G exhibits detectably abnormal convergence behavior in our simulations. When G is incorrectly specified, the absolute parameter change fails to monotonically decrease during EM iterations, providing a diagnostic signal. This aligns with established theory on mixture model misspecification, where incorrect component numbers lead to convergence issues [2]. This built-in diagnostic capability actually makes our approach more transparent than standard neural networks where hyperparameter misspecification may go undetected.
> > > We hope the above clarification addresses your question.
> > >
> > > [1] McLachlan, G. J., & Peel, D. (2000). Finite mixture models. John Wiley & Sons.
> > >
> > > [2] Lindsay, B. G. (1995). Mixture models: Theory, geometry, and applications. IMS.
> > >
> > >
> > > ## Minor Issues
> > >
> > > **E-step terminology.**
> > > **Response**: Thank you for the question. “Posterior probabilities” and “posterior responsibilities” refer to the same quantities; the latter is the term traditionally [1,2] used in the literature for mixture models. To avoid confusion, we have made the terminology consistent by using “posterior probabilities” throughout the revision.
> > > [1] Bishop, C. M., Svensén, M., & Williams, C. K. (1998). GTM: The generative topographic mapping. Neural computation, 10(1), 215-234.
> > > [2] Meeds, E., & Osindero, S. (2005). An alternative infinite mixture of Gaussian process experts. Advances in neural information processing systems, 18.
> > >
> > > **Algorithm 1.**
> > > **Response**: Thank you for pointing this out. Lines 9 and 11 correspond to two variants of ROME-MoE (using $S$ or $(A;S)$ for gating), and in both cases the gating network outputs real-valued logits, with softmax applied once to obtain probabilities. The previous type annotation in lines 3–4 may have implied that the network already outputs normalized probabilities; we have revised this to avoid confusion. Please see the updated Algorithm 1 in the revised manuscript.
> > >
> > > **Term "covariates".**
> > > **Response**: Thank you for the comment. In our synthetic experiments, “covariates” simply refers to the features X = (A, S); the two terms are used interchangeably. We have revised the text to use “features” for clarity.
> > >
> > >
> > > **Table 2 caption.**
> > > **Response**: Thank you for pointing this out. We have corrected the caption in the revised manuscript to consistently reflect the reported metric.

---

### Official Review · Reviewer_qKd7 · 2025-10-31

**Soundness:** 2
**Presentation:** 3
**Contribution:** 2
**Rating:** 4
**Confidence:** 4

**Summary:**

The paper proposes ROME (Robust Mixture Ensemble) to improve algorithmic fairness when harmful disparities arise in latent subgroups rather than predefined demographics. It has two versions: ROME-EM, a finite-mixture linear model trained via EM with a post-hoc distributionally robust (DRO) closed-form aggregation over group predictors; and ROME-MoE, a neural mixture-of-experts that routes by a gating network (which may use sensitive attributes) while restricting experts to non-sensitive features and training with a worst-group objective. The performance of this method is showcased via simulation and three real-world datasets. The approach requires no predefined group labels but does rely on sensitive attributes during training.

**Strengths:**

Fairness with latent structure is important and underserved. The paper argues that discretizing continuous S or predefining groups is limiting and aims to build a model to enforce algorithmic fairness in the presence of latent subgroups. The paper is easy to read and well-structured.

**Weaknesses:**

1. **Theoretical guarantees and fairness notion.** The paper’s theoretical footing, in particular regarding algorithmic fairness, is unclear. It is not specified which formal criterion of algorithmic fairness (e.g., Demographic Parity, Equalized Odds, etc.) is being optimized here. Can the authors provide finite-sample guarantees in a simple linear setting to anchor the claims?

2. **Sensitivity to the number of groups ($G$).** How dependent is performance on knowing or correctly selecting $G$? Some analysis of robustness to misspecification and practical guidance for choosing $G$ would be helpful.

3. **Use of sensitive attributes in gating.** Allowing $S$ in the gating network induces an indirect $S \rightarrow$ prediction pathway via routing, even if experts do not observe $S$. Could this, in practice, violate algorithmic fairness?

4. **Empirical evaluation.**
   - **Baselines.** The current baselines are weak: training on non-sensitive features $A$ alone is known not to ensure fairness when $S$ correlates with $A$. Please include stronger, established fair-learning baselines for a meaningful comparison.
   - **Effect sizes.** Reported improvements over a simple MLP baseline appear small. While they may be significant from the standard error point of view, their practical significance is minimal, especially given the training cost.

**Questions:**

Please see the weakness section.

---

> ### Author Response · Authors · 2025-11-20
>
> Thank you for the thoughtful and constructive feedback. We have carefully revised the manuscript to address each point. Below we provide a point-by-point response. All proposed revisions have been incorporated into the revised manuscript (major changes highlighted in blue and minor edits reflected directly).
>
>
> ## Weakness
>
> **W1. Theoretical guarantees and fairness notion.**
> **Response:** Thank you for raising these points.
>
> The formal fairness criterion we adopt is group-robust fairness, where a model is fair if its predictive performance is stable across (observed or latent) subpopulations—i.e., worst-group risk and group-wise disparities are small.
> Most widely used fairness metrics (e.g., demographic parity, equalized odds, predictive parity, FPR/FNR gaps) are defined only for binary classification and do not extend naturally to continuous outcomes; applying them directly in regression settings produces ill-defined or misleading interpretations [1–3].
> In regression and DRO-based literature, the standard fairness quantities are group-wise losses, including worst-group MSE, risk gaps, and average–worst disparities. These are the metrics used in our real-data experiments. We now explicitly state this fairness notion in Section 2.1 of the revised manuscript.
>
> On the theoretical side, our main contribution is to adapt the closed-form maximin aggregation principle of Wang et al. to the latent-group setting. In the linear model with known groups, their results provide population-level maximin optimality and finite-sample guarantees for the robust aggregation weights. In ROME-EM, we first estimate group-specific predictors via EM and then apply this same aggregation rule to the estimated components; under correct model specification and sufficient sample size, this recovers the same robust predictor asymptotically. We have clarified this connection in the Discussion section of the revised manuscript.
>
> [1] Bird, S., Dudík, M., Edgar, R., Horn, B., Lutz, R., Milan, V., ... & Walker, K. (2020). Fairlearn: A toolkit for assessing and improving fairness in AI
>
> [2] Hong, C., Liu, M., Wojdyla, D. M., Hickey, J., Pencina, M., & Henao, R. (2024). Trans-balance: Reducing demographic disparity for prediction models in the presence of class imbalance. Journal of biomedical informatics, 149, 104532.
>
> [3] Du, W., Xu, D., Wu, X., & Tong, H. (2021). Fairness-aware agnostic federated learning. In Proceedings of the 2021 SIAM International Conference on Data Mining (SDM) (pp. 181-189). Society for Industrial and Applied Mathematics.
>
>
> **W2. Sensitivity to the number of groups G**
> **Response:** Thank you for your comment. We respectfully note that G in our framework is a hyperparameter, and the question of 'robustness to incorrect G' is essentially a model misspecification question, which has been extensively studied in the mixture model literature [1,2].
> For our neural network-based ROME-MoE approach, we conducted sensitivity analysis across $G \in \{2, 3, 4\}$ using Crime data and found that: (1) The fine-tuned value of G indeed achieves the best worst-group performance compared to the two other candidates (2%-10% better), confirming that standard hyperparameter selection via cross-validation can effectively identify optimal G. (2) Performance degrades gracefully with suboptimal G, similar to using suboptimal hyperparameters in standard neural networks.
> For ROME-EM, model misspecification with incorrect G exhibits detectably abnormal convergence behavior in our simulations. When G is incorrectly specified, the absolute parameter change fails to monotonically decrease during EM iterations, providing a diagnostic signal. This aligns with established theory on mixture model misspecification, where incorrect component numbers lead to convergence issues [2]. This built-in diagnostic capability actually makes our approach more transparent than standard neural networks where hyperparameter misspecification may go undetected.
> We hope the above clarification addresses your question.
>
> [1] McLachlan, G. J., & Peel, D. (2000). Finite mixture models. John Wiley & Sons.
>
> [2] Lindsay, B. G. (1995). Mixture models: Theory, geometry, and applications. IMS.

---

> > ### Author Response · Authors · 2025-11-20
> >
> > **W3. Use of sensitive attributes in gating.**
> > **Response:** Thank you for the question. We would like to clarify that in ROME-MoE the experts never receive the sensitive features $S$; only the gating network uses S to infer latent subgroup membership. Thus, $S$ influences which predictor is applied but never affects the functional form of the prediction itself. This routing mechanism does not constitute unfair use of S, because it mirrors the standard practice in latent-group fairness and mixture modeling, where S is used to estimate subgroup structure but is excluded from the final prediction rule.
> > Importantly, users specify which variables are considered fairness-sensitive based on legal, ethical, or institutional guidelines. For instance, if clinical regulations prohibit diagnosis models from depending on ethnicity, then ethnicity belongs to S and is excluded from the experts; if admissions decisions must not depend on gender, then gender is treated similarly.
> > This design is also directly analogous to ROME-EM, where $S$ is used only to estimate mixture memberships. We have clarified this distinction explicitly in Section 3 of the revised manuscript.

---

> > > ### Author Response · Authors · 2025-11-20
> > >
> > > **W4. Empirical evaluation..**
> > > **Response:** Thank you for the question. Please see our response below:
> > >
> > > 1) **Baselines**.
> > >     We acknowledge your concern. The challenge is that most established fairness methods either require known discrete sensitive attributes $S$, or fail to achieve the goal that we use $S$ information for achieving a fair model but avoid making $S$ used in the final prediction model.
> > >     Given this, we have added an oracle-group baseline based on GroupDRO applied to the same MLP architecture as our existing baselines. Here, groups are defined using discretized or intersectional sensitive attributes (e.g., race × gender), which enables us to incorporate a widely used fairness-oriented DRO method even though it assumes observed groups. This provides a more comprehensive comparison and helps isolate the contribution of the robust aggregation mechanism in ROME-MoE.
> > > The details of this new baseline have been added to the Experiment section, and the new results are reported in Tables 2–5 of the revised manuscript. As shown, this baseline is still outperformed by our proposed methods, highlighting the advantages of ROME-MoE.
> > >
> > >
> > > 2) **Effect sizes**.
> > >     We appreciate this important practical consideration.
> > >     First of all, most available fairness benchmarks were designed for binary classification tasks (e.g., COMPAS, Adult Income) where bias is measured through metrics such as demographic parity, equalized odds, predictive parity, or FPR/FNR gaps [1,2,3]. These metrics do not extend naturally to continuous outcomes or multi-class settings, and applying them directly to regression problems can lead to ill-defined or misleading interpretations. For regression tasks, fewer established datasets exist with clear fairness implications. The Crime and Law School datasets we use, while standard, may not fully capture scenarios where latent group robustness would show larger practical benefits. Therefore, we anticipate that our methods will demonstrate broader impact when applied to more suitable datasets.
> > >     Secondly, we'd like to clarify an important misunderstanding about the baseline comparisons. The reviewer notes that 'improvements over a simple MLP baseline appear small,' but this overlooks a crucial distinction in our experimental setup.
> > > The 'Baseline MLP' in our tables uses both A and S features, making it an unfair model that violates the problem constraint (S should not be used for prediction). The only valid comparison is with 'Baseline MLP - Fair,' which uses only A features, matching our methods' constraint. When compared to this appropriate baseline, the improvements are substantial especially for Worst-group $R^2$: This is where the impact is most dramatic. Baseline MLP - Fair often produces negative R² values for worst-performing groups (please see Tables 3-5 in Appendix), indicating predictions worse than using the mean. Our methods consistently achieve positive values of 0.1-0.2, transforming completely failed predictions into meaningful ones.
> > > This improvement from negative to positive  represents a fundamental qualitative change - from a model that fails entirely on certain subpopulations to one that provides reasonable predictions for all groups. This is not a minor improvement but rather the difference between a deployable and non-deployable system in fairness-critical applications.
> > >
> > > [1] Bird, S., Dudík, M., Edgar, R., Horn, B., Lutz, R., Milan, V., ... & Walker, K. (2020). Fairlearn: A toolkit for assessing and improving fairness in AI
> > >
> > > [2] Hong, C., Liu, M., Wojdyla, D. M., Hickey, J., Pencina, M., & Henao, R. (2024). Trans-balance: Reducing demographic disparity for prediction models in the presence of class imbalance. Journal of biomedical informatics, 149, 104532.
> > >
> > > [3] Du, W., Xu, D., Wu, X., & Tong, H. (2021). Fairness-aware agnostic federated learning. In Proceedings of the 2021 SIAM International Conference on Data Mining (SDM) (pp. 181-189). Society for Industrial and Applied Mathematics.

---

> ### Author Response · Authors · 2025-11-20
>
> ## Questions
>
> **Q1.(a-c)**
> **a.** What is the definition of the outcome prediction model mentioned in line 077?
> **b.** What is the difference between $S_{i, \text{mem}}$ and $S_{i, \text{out}}$?
> **c.** How are these two sets of sensitive attributes determined in practice?
>
> **Response**: Thank you for these related questions.
> We distinguish $S_{i,\text{mem}}$ and $S_{i,\text{out}}$ because the latent subgroup structure may depend on certain sensitive attributes, while the final prediction model may not be allowed to use all of them.
> For example, a latent group may depend on age, so including age in $S_{i,\text{mem}}$ can help ROME identify subgroup structure. Whether age is permitted in the final predictive model depends on domain knowledge, institutional policy, and legal regulations:
>
> If age is allowed as a predictive feature, it appears in both $S_{i,\text{mem}}$ and $S_{i,\text{out}}$.
>
> If age is not allowed, it is included only in $S_{i,\text{mem}$ to assist subgroup estimation and is excluded from the final outcome model.
>
> Thus, the “outcome prediction model” referenced at line 077 refers to the final model used in the application, while the allocation of attributes $S_{i,\text{mem}}$ and $S_{i,\text{out}}$ is determined by domain expertise and relevant fairness or compliance requirements. We have clarified this distinction in the revision (Section 2.1).
>
> **Q2.**
>
> **a.** For ROME-EM and ROME-MoE, how sensitive is the method to initialization?
> **Response:** For ROME-EM, the behavior is similar to standard EM for mixture regression: the algorithm is, in principle, sensitive to initialization. In our simulation studies, we found that even when 50–60% of the initial group labels were randomly misassigned, the algorithm still converged, although it may require more iterations.
> For ROME-MoE, we use the standard PyTorch initialization for the MLP-based experts and gating network (i.e., the default initialization of nn.Linear layers). Training proceeds via gradient-based optimization, so the sensitivity resembles that of typical MoE or deep models: different random seeds can lead to slightly different local minima, but we did not observe any catastrophic instability in our experiments.
> We have further clarified these points in the revised manuscript.
>
>
> **b.** How is the number of groups $G$ selected?
> **Response:** In both the ROME-EM and ROME-MoE implementations, the number of groups G is treated as a model-selection hyperparameter. It can be tuned using metrics such as MSE, R^2, worst-group performance, or information criteria (AIC/BIC) in the regression setting, following standard practice in mixture models and MoE architectures. In practice, G is kept small (e.g., G = 2,3,4) because (i) larger values tend to over-partition the data and lead to unstable subgroup estimates, and (ii) the goal is to capture coarse latent heterogeneity rather than fine-grained clustering. Empirically, our results were stable across these reasonable choices of G.
>
>
> **c.** What does “sufficiently large” mean for $n_j$  in the context of convergence?
> **Response**: Thank you for the question. In ROME-EM, “sufficiently large” refers to the usual requirement in mixture regression that each latent group must have enough effective sample size to support a stable weighted least-squares update in the M-step. This aligns with prior work on EM and Monte Carlo EM implementations, which emphasize that reliable convergence requires well-conditioned M-step updates, and that mixture components with very small effective sample sizes tend to produce unstable parameter estimates and poor convergence behavior [1,2]. We have added the corresponding references in the revised manuscript for readers’ convenience.
>
> [1] Ng, S. K., Krishnan, T., & McLachlan, G. J. (2011). The EM algorithm. In Handbook of computational statistics: concepts and methods (pp. 139-172). Berlin, Heidelberg: Springer Berlin Heidelberg.
> [2] Levine, R. A., & Casella, G. (2001). Implementations of the Monte Carlo EM algorithm. Journal of Computational and Graphical Statistics, 10(3), 422-439.

---

> > ### Author Response · Authors · 2025-11-20
> >
> > **Q3. Regaridng the definition of $\mathcal{H}$.**
> > **Response:**  Thank you for the comment.
> >
> > To clarify, in Eq. (7) we follow the commonly used notation in mixture models and distributionally robust optimization [1,2], where the probability simplex is often written as $\mathcal{H} = \left\{ \mathbf{v} \in \Delta^{G} : \|\mathbf{v} - \mathbf{v}_0\|_{2} \leq c \sqrt{G} \right\}$.
> >
> > This set is indeed a (G-1)-dimensional simplex geometrically, since it satisfies one linear constraint, but in computation it is always represented as a $G$-dimensional vector. For this reason—and for consistency with standard practice and our code implementation—we keep the full $G$-dimensional notation.
> >
> > [1] Bühlmann, P., & Meinshausen, N. (2015). Magging: maximin aggregation for inhomogeneous large-scale data. Proceedings of the IEEE, 104(1), 126-135.
> >
> > [2] Wang, Z., Bühlmann, P., & Guo, Z. (2023). Distributionally robust machine learning with multi-source data. arXiv preprint arXiv:2309.02211.
> >
> >
> >
> > **Q4. Could the authors further explain why the constraint set adopted in Eq. (7) clarifies or improves the ROME-EM formulation?**
> > **Response:** Thank you for the question. The constraint set in Eq. (7) plays a stabilizing and interpretable role in the ROME–EM update. Without any restriction, the worst-case weight vector $v$ could put nearly all its mass on a single subgroup, yielding an overly pessimistic objective and causing unstable EM updates (and is inconsistent with the assumption that multiple latent groups are present). The constraint simply limits how far $v$ may deviate from a baseline mixture weights $v_0$, which smooths the updates, prevents extreme adversarial behavior, and improves convergence stability. We have clarified this motivation in the revised manuscript.
> >
> >
> > **Q5. Since subgroup structure is often subtle or latent, how does the method determine which attributes are non-fairness-related (i.e., can be safely used in predictors)?**
> > **Response:** Thank you for the question. As explained in the "Weaknesses" section (W2), this choice is determined by domain knowledge and policy, not by the machine learning or modeling method. Users specify which variables are fairness-related based on legal, ethical, or institutional guidelines. For example, if clinical regulations prohibit diagnostic models from depending on ethnicity, then ethnicity is included in the sensitive set $S$ and excluded from predictors. Similarly, if an admissions model must not depend on gender, then gender is treated as sensitive.
> > Our framework fully accommodates such user-defined choices: $S_{\text{mem}}$ (sensitive attributes) is used only for group membership estimation, while predictors access only the non-sensitive attributes $A$, unless the domain explicitly permits certain sensitive variables to appear in the outcome model. This separation ensures that practitioners can adhere to their domain’s fairness requirements while still leveraging our method to capture latent subgroup structure.
> >
> >
> > **Q6. Are there any detailed analyses about the intersectional unfairness mitigation or detection?**
> > **Response:** Thank you for the question. Yes. In our experiments (Section 4.2, Real Data Analysis), the fairness evaluation already incorporates intersectional structure. For each dataset, we define evaluation groups as either (i) partitions induced by a single sensitive attribute or (ii) the joint combinations of multiple sensitive attributes (e.g., race × gender). All reported worst-group metrics and fairness gaps are computed over these intersectional subgroups. These group definitions appear in Tables 1–5, and the exact sensitive variables used (S) are listed in Appendix B.
> >
> >
> > ## Additional Baseline
> > Beyond the responses above, we have also implemented an oracle DRO baseline per requested by other reviewers. In the revised manuscript, we have added an oracle-group baseline based on GroupDRO applied to the same MLP architecture as our existing baselines. Here, groups are defined using discretized or intersectional sensitive attributes (e.g., race × gender), which enables us to incorporate a widely used fairness-oriented DRO method even though it assumes observed groups. This provides a more comprehensive comparison and helps isolate the contribution of the robust aggregation mechanism in ROME-MoE. The details of this new baseline have been added to the Experiment section, and the new results are reported in Tables 2–5 of the revised manuscript. As shown, this baseline is still outperformed by our proposed methods, highlighting the advantages of ROME-MoE.
> > We believe this additional baseline further demonstrates the value of our approach.

---

### Official Review · Reviewer_8ajM · 2025-11-02

**Soundness:** 3
**Presentation:** 2
**Contribution:** 2
**Rating:** 4
**Confidence:** 3

**Summary:**

In this paper, the authors propose a Robust Mixture Ensemble (ROME) framework to address the problem of group fairness.
Built upon the EM and Mixture-of-Experts (MoE) algorithmic paradigms, the proposed method can automatically identify latent intersectional subgroups within data and employs DRO to minimize the worst-case performance, thereby enhancing fairness.
Experimental results on both synthetic and real-world tabular datasets validate the effectiveness of the proposed approach.

**Strengths:**

- The consideration of intersectional effects arising from combinations of sensitive attributes is both interesting and important.
- The proposed EM- and MoE-based DRO method is well aligned with the formulated fairness problem and theoretically grounded.
- The paper is overall well-organized and easy to follow, with a clear problem formulation and experimental structure.

**Weaknesses:**

- Lack of experimental details.

  For example, the paper does not clearly describe the process of synthetic data generation, nor does it provide sufficient information about the neural network architecture used for the gating network and predictors.

- Fairness metric and definition are insufficiently discussed.

  The performance comparison mainly focuses on subgroup differences under specific evaluation metrics (e.g., MSE), but the scope or interpretation of fairness is not clearly defined.

- Presentation could be further refined.

  Certain explanations lack precision or consistency; please refer to the questions below for clarification suggestions.

- Theoretical analysis needs stronger connections.

  While the theoretical results mainly build upon prior work [1], it would be valuable for the authors to further explain how these existing results specifically motivate or strengthen the current method.

> [1] Distributionally Robust Machine Learning with Multi-Source Data.

**Questions:**

- What is the definition of the outcome prediction model mentioned in line 077?
  What is the difference between $\mathbf{S}\_{i,\text{mem}}$ and $\mathbf{S}_{i,\text{out}}$?
  How are these two sets of sensitive attributes determined in practice?

- For ROME-EM and ROME-MoE, how sensitive is the method to initialization?
  How is the number of groups G selected?
  What does “sufficiently large” mean for $n_j$ in the context of convergence?

- The definition of $\mathcal{H}$ appears to be inaccurate.
  $\mathbf{v}$ should belong to a $(G-1)$-dimensional probability simplex $\Delta^{G-1}$.

- Could the authors further explain why the constraint set adopted in Eq. (7) clarifies or improves the ROME-EM formulation?

- Since subgroup structure is often subtle or latent, how does the method determine which attributes are non-fairness-related (i.e., can be safely used in predictors)?

- Are there any detailed analyses about the intersectional unfairness mitigation or detection?

---

> ### Author Response · Authors · 2025-11-20
>
> We sincerely thank you for your thoughtful feedback. We are encouraged that you found the paper “well-organized and easy to follow” and that the core method is “well aligned with the formulated fairness problem and theoretically grounded.”
> Below, we provide a point-by-point response to further clarify the work and address all concerns. The proposed revisions and additions have been incorporated into the revised manuscript (major changes marked in blue).
>
>
> ## Weakness
>
> **W1. Lack of experimental details.** For example, the paper does not clearly describe the process of synthetic data generation, nor does it provide sufficient information about the neural network architecture used for the gating network and predictors.
>
> **Response:** Thank you for this comment. Because of the page limit at submission time, detailed simulation settings were originally placed in Appendix A. In the revision, we have further clarified the following:
> (a) **Synthetic data generation.**
> We now explicitly state in Section 4.1 that the synthetic data are generated in R according to the model in the main text, with all parameter values listed in Appendix A.1. We also added a footnote pointing readers to the anonymous repository, where the full R implementation (including the script "demo.R") is available for reproducibility.
> (b) **Neural network architecture.**
> All MoE-based models use standard two-layer MLPs for both experts and the gating network. We added a concise description of the inputs to the experts and gating net, the hidden sizes / activations, and the output structure in Appendix D.
>
>
> **W2. Fairness metric and definition are insufficiently discussed.** The performance comparison mainly focuses on subgroup differences under specific evaluation metrics (e.g., MSE), but the scope or interpretation of fairness is not clearly defined.
>
> **Response:** Thank you for raising this point. In this work we focus on algorithmic (group) fairness. The formal fairness notion we adopt is that a predictor is fair if its predictive performance is stable across subpopulations defined by sensitive attributes or latent groups—that is, if worst-group risk and group-wise disparities are small.
> Most widely used fairness metrics in the literature—such as demographic parity, equalized odds, predictive parity, or FPR/FNR gaps [1–3]—are designed for binary classification and do not extend naturally to continuous outcomes or multi-class responses. Applying them directly in regression settings can lead to ill-defined or misleading interpretations; this limitation has been discussed in prior work on group robustness and fair regression [1–3].
> In regression and DRO settings, the standard fairness criteria are group-wise risk measures, such as worst-group MSE, average–worst group gaps, and risk disparities. These are exactly the quantities we report in our real-data experiments (Section 4.2), and we now explicitly state this fairness notion in Section 2.1 of the revised manuscript.
>
> [1] Bird, S., Dudík, M., Edgar, R., Horn, B., Lutz, R., Milan, V., ... & Walker, K. (2020). Fairlearn: A toolkit for assessing and improving fairness in AI
>
> [2] Hong, C., Liu, M., Wojdyla, D. M., Hickey, J., Pencina, M., & Henao, R. (2024). Trans-balance: Reducing demographic disparity for prediction models in the presence of class imbalance. Journal of biomedical informatics, 149, 104532.
>
> [3] Du, W., Xu, D., Wu, X., & Tong, H. (2021). Fairness-aware agnostic federated learning. In Proceedings of the 2021 SIAM International Conference on Data Mining (SDM) (pp. 181-189). Society for Industrial and Applied Mathematics.
>
> **W3. Presentation could be further refined.** Certain explanations lack precision or consistency; please refer to the questions below for clarification suggestions.
>
> **Response:** Thank you for pointing this out. Please see our responses in the **Questions** section, where we clarify the specific notational and wording issues.

---

> > ### Author Response · Authors · 2025-11-20
> >
> > **W4. Theoretical analysis needs stronger connections.**
> > While the theoretical results mainly build upon prior work, it would be valuable for the authors to further explain how these existing results specifically motivate or strengthen the current method.
> >
> > **Response:** hank you for this important point. Our theoretical analysis extends Wang et al. in ways that directly motivate our method design:
> >
> > For ROME-EM: Wang et al.'s closed-form maximin solution assumes known groups. We extend this to the latent-group setting by (i) using EM to learn group-specific predictors from unlabeled data, and then (ii) applying their DRO aggregation formula to these learned components. This yields a principled combination rule and interpretable weights that quantify each latent group’s vulnerability under distribution shift.
> > For ROME-MoE: We operationalize the same maximin principle in a neural architecture by treating experts as group-specific predictors and using a DRO-inspired loss that combines average and worst-group errors. This directly transfers the linear maximin objective to the nonlinear MoE setting.
> > In short, DRO theory provides the aggregation principle (how to robustly combine group predictors), and our contribution is to operationalize this principle when groups are latent—via EM for linear models and MoE architectures for neural models. We have clarified these connections in Section 2.3 and at the beginning of Section 3 in the revised manuscript.

---

### Meta-Review · Area_Chair_RKEe · 2025-12-09

**Summary:**

This paper proposes a method designed to mitigate algorithmic discrimination when disparities arise within latent subgroups rather than predefined demographic groups. It introduces two implementations: a finite-mixture linear model trained via EM and a neural mixture-of-experts model with a gating network. Fairness with respect to latent groups is an important and underexplored topic, and the paper takes an interesting step towards addressing this gap. Reviewers pointed out several weaknesses. the authors actively participated in the rebuttal period and revised the paper.

**Reviewer Concerns:**

Many comments have been addressed and notations have been clarified. However, there are two most important points which are still not sufficiently addressed in my opinion.

(1) insufficient discussion of the selected fairness metric.

The paper focuses on worst-group risk and group-wise disparities, but the motivation for using these metrics and their relationship to other group fairness metrics is underdeveloped. This gap is particularly significant given known trade-offs between fairness measures. The paper should clarify the advantages and limitations of the chosen metric and specify the applications where it is most appropriate. Additionally, the claim that “most widely used fairness metrics are designed for binary classification and do not extend naturally to continuous outcomes or multi-class responses” is not fully accurate. There is a growing body of work extending classical group fairness notions to multi-class settings and regression tasks.

(2) lack of baseline comparisons.

Reviewers pointed out that the paper lacks "stronger, established fair-learning baselines" in the experiment section. The authors responded that most existing fairness methods require known sensitive attributes, whereas this work focuses on latent subgroups. While this is a valid point, the paper does not sufficiently motivate why latent subgroups are practically important. Incorporating real-world examples or datasets could strengthen this justification. Moreover, there is a line of work on achieving group fairness when sensitive attributes are partially missing or unobserved, and comparisons with those methods would also be valuable.

**Reviewer Scores:**

All reviewers pointed out that experiment is a main weakness of the paper. The authors only added an oracle baseline. Additionally, the clarification of the fairness metric considered in this paper is not sufficiently motivated. I would guess the reviewers might keep the scores the same.

---

### Decision · Program_Chairs · 2026-01-26

Reject